# Repurposing Marigold for Zero-Shot Metric Depth Estimation via Defocus Blur Cues

**Chinmay Talegaonkar**,* **Nikhil Gandudi Suresh, Zachary Novack, Yash Belhe,**
**Priyanka Nagasamudra, Nicholas Antipa**
University of California San Diego

## Abstract

Recent monocular metric depth estimation (MMDE) methods have made notable progress towards zero-shot generalization. However, they still exhibit a significant performance drop on out-of-distribution datasets. We address this limitation by injecting defocus blur cues at inference time into Marigold, a *pretrained* diffusion model for zero-shot, scale-invariant monocular depth estimation (MDE). Our method effectively turns Marigold into a metric depth predictor in a training-free manner. To incorporate defocus cues, we capture two images with a small and a large aperture from the same viewpoint. To recover metric depth, we then optimize the metric depth scaling parameters and the noise latents of Marigold at inference time using gradients from a loss function based on the defocus-blur image formation model. We compare our method against existing state-of-the-art zero-shot MMDE methods on a self-collected real dataset, showing quantitative and qualitative improvements. Our implementation is available at `https://github.com/chinmay0301ucsd/DiffusionCam`.

## 1 Introduction

Estimating metric depth from a single camera viewpoint is a central problem in computer vision with numerous downstream applications, including 3D reconstruction [24], autonomous driving [50], and endoscopy [31]. This task, known as *monocular metric depth estimation* (MMDE), is fundamentally ill-posed due to inherent depth-scale ambiguity [46]. Multi-view methods [65] avoid this ambiguity but are often expensive and impractical in settings like endoscopy or microscopy. Training data-driven MMDE methods is challenging, as it requires accounting for a diverse set of camera parameters and metric depth scales. As a result, existing MMDE models struggle in zero-shot settings, i.e., they generalize poorly to unseen datasets. Recent advances in zero-shot MMDE [75] have demonstrated improved generalization, but there is still a considerable performance drop on unseen datasets.

In contrast to MMDE, monocular *relative* depth estimation (MDE) methods recover a relative depth map, factoring out the physical depth scale. This enables using large-scale datasets with diverse depth ranges [41] for training data-driven MDE methods. As a result, MDE methods achieve better zero-shot generalization at significantly lower training cost than MMDE methods, as shown by recent results [70, 71, 26]. However, despite favorable performance on benchmarks, the absence of metric scale in MDE outputs precludes their applicability in downstream tasks requiring absolute depth.

Existing data-driven MMDE methods commonly suffer from two failure modes: undesirable coupling between image texture and depth predictions (fig. 3), and inaccurate estimation of the scene's physical scale (see fig. 4). The first issue of *texture coupling* also affects MDE methods (see fig. 3) unless explicitly mitigated through complex training procedures that account for texture variation [22, 55].

---

*Corresponding author: `ctalegaonkar@ucsd.edu`

39th Conference on Neural Information Processing Systems (NeurIPS 2025).

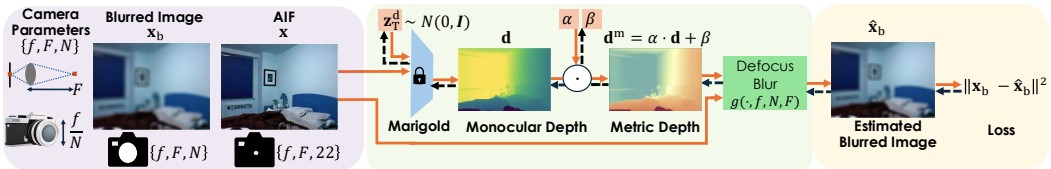

Figure 1: **Method overview.** We capture two images (same viewpoint) from a camera with focal length $f$ and focused at a distance $F$: an all-in-focus (AIF) image $\mathbf{x}$ (F-stop: $N = 22$) and a blurred image $\mathbf{x}_b$ (F-stop: $N < 22$). Using the AIF $\mathbf{x}$ and an initial *learnable* noise vector $\mathbf{z}_T^{(\mathbf{d})}$, Marigold predicts the relative depth $\mathbf{d}$. We then affine transform $\mathbf{d}$ with *learnable* parameters $(\alpha, \beta)$, obtaining the metric depth $\mathbf{d}^m$. Given the AIF $\mathbf{x}$, depth $\mathbf{d}^m$, and camera parameters $(f, F, N)$, we synthesize a blurred image $\hat{\mathbf{x}}_b$ using the defocus blur forward model. To update the learnable parameters, we compute their gradients w.r.t the L2 loss between $\hat{\mathbf{x}}_b$ and $\mathbf{x}_b$.

We demonstrate that incorporating camera physics, particularly defocus blur, with a data-driven MDE model at inference time can effectively address both failure modes *without any re-training*. We use Marigold [26], a diffusion-based MDE model, as it enables backpropagation-based inference time optimization using defocus cues, while staying on its learned manifold of plausible depth maps. To provide these defocus blur cues, we capture two images from a single viewpoint: a small-aperture all-in-focus (AIF) image, which is given as input to Marigold, and a large-aperture image that provides physical cues for metric depth. Our approach requires only a variable aperture camera, such as a DSLR, and knowledge of the lens focal length, focus distance, and F-stop, which are readily available from image metadata; this avoids the need for extrinsic calibration required by multiview setups.

Defocus blur is known to provide coarse metric depth cues in monocular settings [33, 59, 69], but has rarely been explored alongside recent data-driven advances in MMDE. We show in our work that revisiting this classical cue and integrating it with modern relative depth methods like Marigold improves metric depth estimation performance, without any retraining. While most depth from defocus methods use multiple measurements [21, 30, 63], we require only two. Prior work has also explored other hardware novelties to encode depth cues, such as dual pixel cameras [19, 68] and coded apertures [28]. However, these depth cues are coarse, and training MMDE models for these systems is limited by the lack of datasets captured with a sufficient camera parameter diversity. This motivates the need for an approach that combines strong data-driven pre-trained MDE/MMDE models trained on existing RGB datasets with physics-based cues from non-pinhole cameras.

To this end, our **contributions** are as follows: We formulate MMDE as an inverse problem under the defocus blur image formation model. Using inference-time optimization, we recover the metric depth scale and correct for texture-depth coupling – without retraining Marigold. We devise a hardware setup consisting of a rigidly coupled DSLR (RGB camera) and an Intel RealSense depth camera to capture RGB images and the ground truth metric depth. We collect a dataset of 7 diverse real-world indoor scenes, captured at different defocus blur levels. We compare our method on this dataset with current MMDE methods and demonstrate quantitative improvements on real data.

## 2 Related Work

**Zero-shot depth models** Recent approaches for Monocular (relative) Depth Estimation (MDE) and Monocular Metric Depth Estimation (MMDE) can be classified as discriminative or generative. Discriminative approaches largely rely on vision transformer-based architectures [7, 5, 6, 74, 39] trained on large-scale datasets. Transformer-based methods have been more successful for MDE [70, 71] compared to MMDE accuracy-wise, as MMDE is a more ill-posed task than MDE. These approaches have very low inference times, but for MMDE, the performance degrades on out-of-distribution test scenes[2]. Most of these methods are purely data-driven and ignore depth-based visual effects in images such as defocus cues. It is also difficult to incorporate physics-based depth refinement in these methods at test time without any re-training [73]. Since we use a generative MDE, our approach can incorporate physical cues at test time in a training-free manner.

---

[2]Refer to [75] for a detailed survey on MMDE methods.

Current state-of-the-art *generative* MDE/MMDE methods [47, 13] are predominantly diffusion-based. Several previous methods [13, 47, 32] incorporate diffusion-based depth denoising in their pipelines, achieving highly detailed depth maps – but are not zero-shot. [48] achieves zero-shot MMDE with a diffusion-based approach by incorporating diverse field of view (FOV) augmentations in training, but it is not open source and lags behind transformer-based methods in performance. Marigold [26] is trained by fine-tuning Stable Diffusion-v2 on synthetic depth data. It achieves high-quality zero-shot MDE and supports test time refinement, but is not applicable natively for MMDE. Our approach uses defocus cues to refine the relative depth predictions from Marigold (or similar methods [16]), enabling its application to MMDE. Prior work [60] also proposes a similar strategy for dense MMDE from a sparse metric depth map using Marigold and test time optimization. In contrast, our method does not require a sparse depth map as input, and solely relies on RGB images and a priori known scene bounds. Using defocus blur cues, our method resolves inaccuracies in monocular depth while also estimating the global scaling parameters for metric depth.

**Diffusion model priors for inverse problems**  We frame MMDE as an inverse problem under the defocus blur image formation model. This framing closely relates to the recent work on solving linear inverse problems using *pre-trained* diffusion models [12]. [9, 11, 54, 10] incorporate the forward model constraints while sampling pixel-space diffusion models pre-trained on smaller datasets. As a result, these methods require many steps while sampling the diffusion model and have limited generalizability as priors. [53, 44] use latent variable diffusion models (LDMs) as priors, resulting in better in-the-wild generalizability. We use Marigold as the LDM, but instead of incorporating the defocus forward model during sampling, we optimize the latent noise vector based on the error between the observed and predicted image using the forward model. Our approach is inspired by recent methods [37, 36, 77, 61, 17] which uses noise optimization in conjunction with a differentiable auxiliary guidance loss to improve the sampling quality of the diffusion model based on text input. While these methods use trained models as differentiable proxies for guidance, we use a physics-based imaging forward model. [34] also uses a physics-based forward model with a diffusion prior, but requires re-training the diffusion prior from scratch.

**Depth estimation from passive camera physics cues**  A substantial body of research on MMDE leverages camera physics, including methods like depth-from-defocus (DfD), -[64, 20, 58, 2], phase/aperture masks [28, 78, 76, 3, 67], and dual pixel sensors [18, 68, 1]. Classical approaches produce very coarse depth maps. Most DfD methods need a well-aligned [66] multi-image focal stack [29, 57], to achieve good depth quality. Optical mask-based methods pose a harder inverse problem of jointly estimating both AIF and the depth map. We capture only 2 images at the same focus distance and different apertures, requiring no image alignment and AIF estimation. This simplifies depth refinement with minimal added capture time. Previous work has also used variable apertures for classical [15] and learning-based [56] depth estimation methods, but it is not zero-shot. While learning-based approaches [19, 8, 66, 18] help improve the depth map quality for these methods, they don't generalize well to out-of-distribution scenes. Dual pixel-based methods are popular for phone cameras, [18, 38], but are tied to the specific camera architecture. While we show results with a standard DSLR sensor, our approach can be adapted to dual-pixel-style camera architectures as well.

## 3  Preliminaries

**Diffusion-based monocular depth estimation**  Our approach is built on top of Marigold [26], which is a monocular depth estimator trained by fine-tuning the denoising U-Net of StableDiffusion-v2 (SDv2) [43] on synthetic depth data. Marigold allows sampling the conditional distribution of the monocular depth given an input image, $p(\mathbf{d}_0|\mathbf{x})$. In particular, Marigold attempts to generate clean monocular depth maps, $\mathbf{d}_0$ given a clean input image $\mathbf{x}$ by first sampling $\mathbf{d}_T \approx \mathcal{N}(0, \boldsymbol{I})$ from an i.i.d Gaussian distribution, and then iteratively denoising it (where each intermediate step is denoted by $\mathbf{d}_t$) according to a fixed noise schedule with parameters $\alpha_t, \sigma_t$. As SDv2 is a *latent*-diffusion model, the entire generative process happens on encoded latent depth maps $\mathbf{z}_0^{(\mathbf{d})}$ with latent images $\mathbf{z}^{(\mathbf{x})}$ as conditioning. Training Marigold[3] (which we denote as $\hat{\mathbf{x}}_\phi(\cdot)$) involves using the standard denoising loss common in image diffusion works, but on depth maps rather than images, and passes the image

---

[3]While technically SDv2 is trained in $\epsilon$-prediction mode rather than $\mathbf{x}$-prediction, we use $\mathbf{x}$-prediction given their mathematical equivalence and alignment with Marigold-LCM, which uses $\mathbf{x}$-prediction.

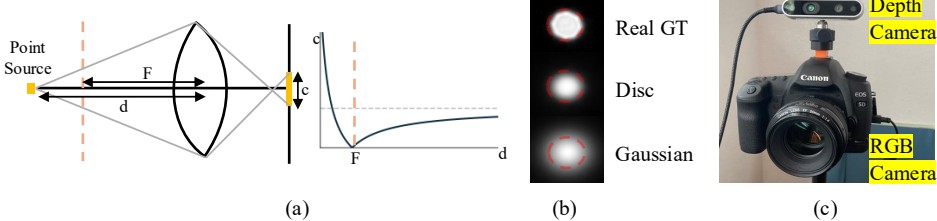

(a)           (b)           (c)

Figure 2: **Comparing simulated PSFs with the PSF captured from our camera setup.** (a) A point source placed $d$ distance away from a thin lens focused at a focus distance $F$ produces a blurred image (PSF) with a diameter $c$, also known as the circle of confusion. The variation of $c$ with source distance $d$ is shown in the plot. (b) The Disc approximation to the camera PSF lies roughly within the same bounds (dotted red circle) as the PSF captured from the RGB camera (Real GT) in (c). The Gaussian PSF significantly exceeds the bounds. Slight differences between the real and Disc PSF stem from the octagonal aperture and diffraction ignored in our model. (c) We rigidly mount an Intel RealSense on a DSLR to capture ground truth depth, and calibrate both cameras to align predicted depth from the RGB image with the ground truth depth for evaluation.

input into the model as well through channel-wise concatenation:

$$\mathbb{E}_{\mathbf{z}_0^{(\mathbf{d})}, \mathbf{z}^{(\mathbf{x})} \sim \mathcal{D}, t \sim p(t), \epsilon \sim \mathcal{N}(0, \mathbf{I})}[w(t) \|\mathbf{z}_0^{(\mathbf{d})} - \hat{\mathbf{x}}_{\boldsymbol{\phi}}(\alpha_t \mathbf{z}_0^{(\mathbf{d})} + \sigma_t \epsilon, \mathbf{z}^{(\mathbf{x})}, t)\|_2^2], \tag{1}$$

where $w(t)$ is a time-dependent weighting function. At inference time, $\mathbf{x}$ is first mapped to a lower dimensional latent vector $\mathbf{z}^{(\mathbf{x})} = \mathcal{E}(\mathbf{x})$ through VAE encoder $\mathcal{E}$ in SDv2. The inference process starts with sampling a noisy latent depth vector $\mathbf{z}_T^{(\mathbf{d})} \sim \mathcal{N}(0, \mathbf{I})$, which is iteratively refined by applying the denoiser $\hat{\mathbf{x}}_{\boldsymbol{\phi}}(\mathbf{z}_t^{(\mathbf{d})}, \mathbf{z}^{(\mathbf{x})}, t)$ to obtain $\mathbf{z}_0^{(\mathbf{d})}$ over $T$ sampling steps. $\mathbf{z}_0^{(\mathbf{d})}$ is then decoded through the SD decoder $\mathcal{D}$ to produce the output depth map $\mathbf{d} = \mathcal{D}(\mathbf{z}_0^{(\mathbf{d})})$. Marigold assumes both $\mathbf{z}^{(\mathbf{x})}, \mathbf{z}^{(\mathbf{d})} \in \mathbb{R}^M$. For faster inference, we use the latent consistency model version of Marigold, Marigold-LCM, and obtain $\mathbf{z}_0^{(\mathbf{d})}$ with a single inference step from $\mathbf{z}_T^{(\mathbf{d})}$. Monocular depth $\mathbf{d}$ can be mapped to metric depth $\mathbf{d}^{\mathrm{m}}$ by an affine transform; more details in section 4.

**Modeling defocus cues**     Under the thin lens camera assumption, defocus blur manifests as a uniform blur kernel, with a depth-dependent diameter. The blur kernel diameter for a point source placed at a distance $d$ away from the camera is determined by the circle of confusion (CoC) [40] equation

$$c(d) = \frac{f^2}{N} \frac{|d - F|}{d(F - f)s}, \tag{2}$$

where the focal length $f$, focus distance $F$, and F-stop $N$ are camera parameters that are known from the image EXIF data. $s$ denotes the pixel size in physical units (m). The CoC defines a depth-dependent point spread function (PSF) $h(i, j \mid u, v, d)$ that predicts the response at pixel coordinate $(i, j)$ from a point source at lateral coordinate $(u, v)$ and depth $d$ under defocus blur. Note that $(u, v)$ can represent depth-normalized pixel coordinates under an ideal pinhole projection, allowing us to write $d = \mathbf{d}^{\mathrm{m}}[u, v]$ (valid for general non-volumetric scenes that assume a single depth value per coordinate). We assume that the PSF is shift-invariant for a given depth $d$, i.e. $h(i, j \mid u, v, d) = h(i - u, j - v \mid 0, 0, \mathbf{d}^{\mathrm{m}}[u, v]) = h(i - u, j - v \mid \mathbf{d}^{\mathrm{m}}[u, v])$, which is simply the on-axis PSF, modeled as a disc (assuming circular aperture) with radius given by CoC equations, translated to be centered at $(i, j)$. To ensure smooth optimization, we include a linear fall-off at the boundary of the discontinuous disc kernel, similar to [62]. The PSF can then be expressed as

$$h(i, j \mid u, v, d) = \frac{\widehat{W}(i, j \mid u, v, d)}{\sum_{i,j} \widehat{W}(i, j \mid u, v, d)}, \text{where} \tag{3}$$

$$\widehat{W}(i, j \mid u, v, d) = \begin{cases} 1, & m \le \frac{c(d) - 1}{2} \\ \frac{c(d) + 1}{2} - m, & \frac{c(d) - 1}{2} < m \le \frac{c(d) + 1}{2} \\ 0, & \frac{c(d) + 1}{2} > m \end{cases}, \tag{4}$$

$$m = \sqrt{(i - u)^2 + (j - v)^2}. \tag{5}$$

We assume the PSF to be normalized and explicitly account for exposure and energy balancing during image capture and processing (see section 4). While the above PSF can also be approximated as an isotropic 2D Gaussian [20], we opt for the disc parameterization in eq. (5) similar to [62, 51], as it better approximates the PSF of the real camera compared to the Gaussian approximation in [20], as shown in fig. 2. We can see from eq. (2) that an image captured with a very small aperture (high $N$) would have negligible defocus blur due to very small CoC values. Such an image is referred to as the all-in-focus (AIF) image, $\mathbf{x}$. Given the AIF $\mathbf{x}$, the blurred image $\mathbf{x}_b$ can be approximated (neglecting occlusion) as a spatially varying convolution between $h$ and $\mathbf{x}$,

$$\mathbf{x}_b(i,j) = \iint \mathbf{x}(u,v) \cdot h(i-u, j-v; \mathbf{d}^m[u,v]) du dv \tag{6}$$

For simplicity, we denote the above image formation forward model as

$$\mathbf{x}_b = g(\mathbf{x}, \mathbf{d}^m, f, F, N). \tag{7}$$

The AIF image $\mathbf{x}$, captured at a high F-stop, serves as the blur-free input for both Marigold and the camera blur model eq. (7), while the low F-stop image $\mathbf{x}_b$ provides defocus cues for MMDE.

**Inference-time optimization**   Without loss of generality, any generative model (GAN, Diffusion or flow-based) $\phi : \mathbb{R}^M \to \mathbb{R}^N$ can be construed as a mechanism to map a simple probability distribution, such as an i.i.d. Gaussian distribution, $\varepsilon \sim \mathcal{N}(0, \boldsymbol{I}) \in \mathbb{R}^M$ to a non-linear $N$-dimensional manifold, such as images or audio, through a differentiable generative process $x = \phi(\varepsilon)$. Inference time optimization [37] refers to manipulating the generation process by updating the *initial noise* $\varepsilon$ based on gradients from a differentiable loss function $\mathcal{L}(x)$ on the generated sample $x$,

$$\epsilon \to \epsilon - \nabla_\epsilon \mathcal{L}(x). \tag{8}$$

To further ensure that $\epsilon$ still lies close to the Gaussian manifold after the gradient updates, $\varepsilon$ can be rescaled to have a L2 norm of $\sqrt{M}$ as in [45], which is a valid approximation for samples drawn from a high dimensional Gaussian distribution as per the Gaussian annulus theorem [4]. This holds for most generative models, as the initial noise vectors are typically high-dimensional ($M > 50$). In our case, $\varepsilon$ corresponds to the noise latent $\mathbf{z}_T^{(\mathbf{d})}$ in Marigold, which we optimize using a loss function (eq. (11)) governed by the defocus blur forward model eq. (7).

## 4   Method

We capture two images per scene: $\mathbf{x}$, with F-stop ($N_{\text{aif}} = 22$) and exposure time $t_{\text{aif}}$, serves as the blur-free all-in-focus (AIF) image. A second image, $\mathbf{x}_b$, is captured at a lower F-stop ($N_b = 8$) with exposure time $t_b$, thereby providing strong depth-varying defocus cues. The forward model in eq. (7) assumes radiometrically linear images (no gamma correction or non-linear processing) and energy constancy between the AIF and blurred images. We use raw images to satisfy these assumptions. Since total captured energy scales with exposure time ($t$) and aperture area ($\propto (f/N)^2$) [25], we scale $\mathbf{x}_b$ by the factor $\frac{t_{\text{aif}}}{t_b} \cdot \frac{N_b^2}{N_{\text{aif}}^2}$ to match the energy in $\mathbf{x}$. Note that we vary the exposure time to ensure well-exposed measurements across F-stop settings, while fixing the camera gain.

We frame metric depth estimation as an inverse problem, with the defocus blur image formation process in eq. (6) as the forward model, and Marigold as the monocular depth prior. To obtain scale-invariant monocular depth $\mathbf{d} \in [0, 1]$, we use Marigold-LCM, which takes in as input the AIF $\mathbf{x}$ (encoded to $\mathbf{z}^{(\mathbf{x})}$), and a learnable depth latent vector $\mathbf{z}_T^{(\mathbf{d})} \sim \mathcal{N}(0, \boldsymbol{I})$. A single inference step of Marigold-LCM gives us the denoised depth latent $\mathbf{z}_0^{(\mathbf{d})} = \hat{\mathbf{x}}_\phi \left( \mathbf{z}_T^{(\mathbf{d})}, \mathbf{z}^{(\mathbf{x})}, 1 \right)$, which is decoded to monocular depth $\mathbf{d} = \mathcal{D}(\mathbf{z}_0^{(\mathbf{d})})$. The predicted monocular depth $\mathbf{d} \in [0, 1]$ is then mapped to metric depth by affine transforming $\mathbf{d}$ with a learnable metric scale ($\alpha$) and offset ($\beta$) per scene, $\mathbf{d}^m = \alpha \cdot \mathbf{d} + \beta$. To ensure that $\alpha, \beta$ remain bounded and differentiable, we parameterize them as $\alpha = s_{\max} \cdot \sigma(a)$ and $\beta = s_{\min} \cdot \sigma(b)$, where $\sigma(\cdot)$ is the sigmoid function; $a, b$ are unconstrained learnable parameters initialized to 0, and $s_{\max}$ and $s_{\min}$ are the upper and lower scene depth bounds, respectively, which we assume are known a priori (valid for indoor scenes). To summarize, the metric

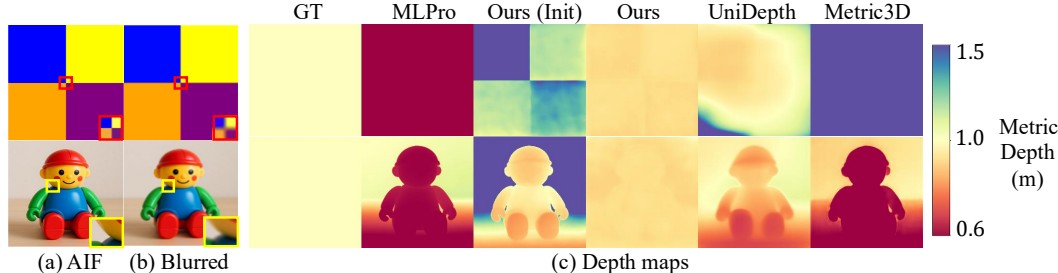

|  |  | GT | MLPro | Ours (Init) | Ours | UniDepth | Metric3D |
| --- | --- | --- | --- | --- | --- | --- | --- |

(a) AIF (b) Blurred (c) Depth maps

Figure 3: **Correcting texture-depth coupling.** We assess MMDE performance on textured fronto-parallel 2D planes with constant ground truth depths (GT). Using an all-in-focus (a) and blurred image (zoom into insets) (b), our method (RMSE: 0.01) recovers the correct depth maps (c) for the two textured planes. We resolve the texture coupling in the Marigold prediction (Ours Init) and recover the correct metric scale. Competing methods (RMSE: 0.2-0.5) fail to predict both the constant relative depth map (except MLPro and Metric3D in row 1) and the correct scale.

depth ($\mathbf{d}^{\mathbf{m}}$) can be expressed using the optimizable parameters $a, b, \mathbf{z}_T^{(\mathbf{d})}$ as:

$$\mathbf{d}^{\mathbf{m}} = s_{\max} \cdot \sigma(a) \cdot \mathcal{D}\left(\hat{\mathbf{x}}_\phi\left(\mathbf{z}_T^{(\mathbf{d})}, \mathbf{z}^{(\mathbf{x})}, 1\right)\right) + s_{\min} \cdot \sigma(b)$$
$$:= y\left(a, b, \mathbf{z}_T^{(\mathbf{d})}\right). \tag{9}$$

The optimized metric depth $\widehat{\mathbf{d}}^m$ can then be recovered by solving:

$$\widehat{\mathbf{d}}^m = \underset{\mathbf{d}^{\mathbf{m}} = y\left(a, b, \mathbf{z}_T^{(\mathbf{d})}\right)}{\arg\min} \quad ||\mathbf{x}_{\mathbf{b}} - g(\mathbf{x}, \mathbf{d}^{\mathbf{m}}, f, F, N)||_2^2 \tag{10}$$

$$\text{subject to} \quad \left\|\mathbf{z}_T^{(\mathbf{d})}\right\|_2 = \sqrt{M}, \tag{11}$$

where $g(\cdot)$ denotes the defocus blur forward model (eq. (7)), $\mathbf{x}_{\mathbf{b}}$ and $\mathbf{x}$ are the captured blurred and AIF images, respectively. By optimizing the learnable parameters $a, b, \mathbf{z}_T^{(\mathbf{d})}$, we incorporate defocus blur cues for both correct metric scale recovery ($a, b$) and refining the initial depth estimate by Marigold ($\mathbf{z}_T^{(\mathbf{d})}$). See fig. 1 for the overview of our method.

**Motivating Synthetic Toy Examples** We demonstrate that our approach resolves texture-depth coupling, accurately recovering metric depth in a synthetic scene with a textured plane at constant depth. While such a plane may seem simple, distinguishing it from a flat 2D image/poster or an actual 3D scene is challenging when viewed from a single viewpoint. Data-driven methods are biased towards predicting depths that reflect surface variations even in the absence of true depth changes, i.e., their outputs are strongly *texture coupled*. However, supplementing the AIF image with a simulated blurred image provides defocus cues that help our method disambiguate a flat poster from a 3D scene, as demonstrated in fig. 3 using toy examples of textured planes with constant depth. While learning-based methods and initial Marigold outputs suffer from texture coupling and scale errors, our method corrects both, producing accurate, constant-depth maps that outperform all baselines.

**Accelerating inference** We use the distilled latent consistency model version of Marigold (Marigold-LCM) to reduce the number of sampling steps significantly. We observe that a single sampling step suffices for our case, which significantly speeds up inference-time optimization [36, 14] relative to the normal 20-50 inference steps [37, 61] that Marigold [60] uses. We show an ablation study with more sampling steps in supplement.E. We also implement custom CUDA kernels for the Disc-PSF forward model. This provides a 2.5x speed up over the PyTorch implementation provided by [62] while being more memory efficient, allowing our method to scale to higher-resolution images. Using a single sampling step of Marigold-LCM allows us to compute gradients w.r.t $\mathbf{z}_T^{(\mathbf{d})}$ without gradient checkpointing as previously done in [37].

**Optimization details** We run the optimization for 200 iterations, which takes roughly 3.5-4 minutes on an NVIDIA A-40 GPU with peak memory usage of 15 GB. We use the Adam optimizer with a

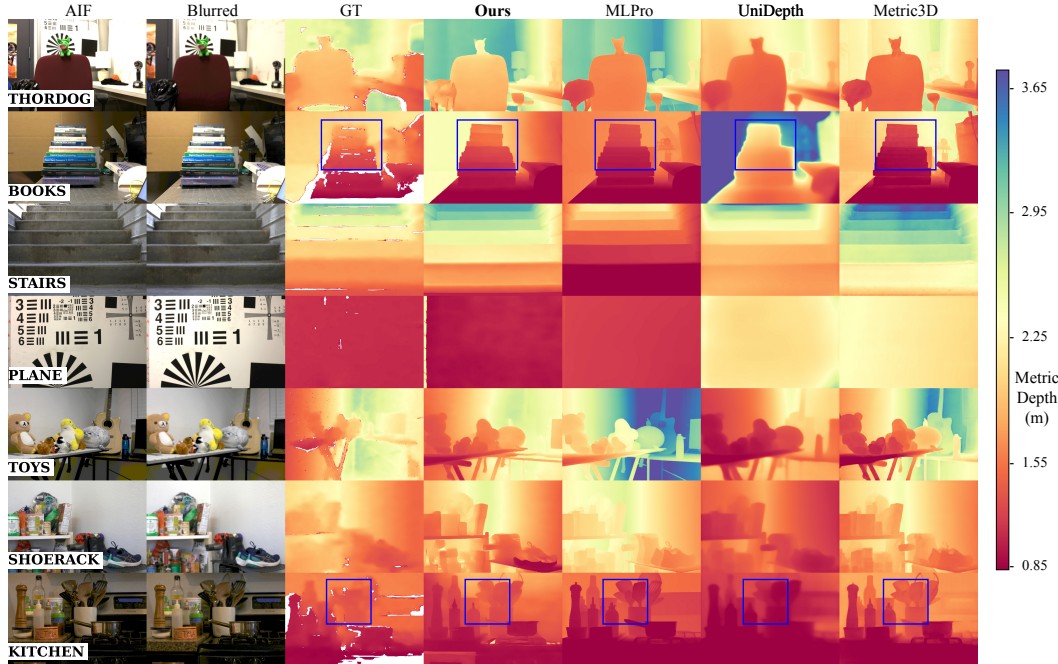

Figure 4: **Comparisons on our collected dataset.** Our method consistently estimates accurate metric depth across all the scenes. We also observe better relative depth recovery due to leveraging defocus cues (zoom in 4x on blurred) in some regions (blue boxes, STAIRS). While the competing methods perform comparably to ours in some cases (MLPro:PLANE, KITCHEN, UniDepth:STAIRS, Metric3D:THORDOG, BOOKS), they struggle with the rest of the scenes due to incorrect relative depth (TOYS, BOOKS) and metric scale (PLANE) recovery. recovers sharp details but fails at metric scale and relative depth accuracy for many of the scenes. Since the RealSense has a wider FOV than the DSLR, we show a roughly aligned crop of the GT depth for comparison.

learning rate of $1.5 \times 10^{-3}$ for $\mathbf{z}_T^{(\mathbf{d})}$, $5 \times 10^{-3}$ for $(a, b)$, and default values for optimizer parameters. Note that we use the *same* scene bounds $s_{\min} = 1.49$, $s_{\max} = 3.5$ for all the real scenes in our dataset. These values represent a conservative upper bound on the potential maximum scale and offset in the real dataset. Please see supplement.A for more details.

## 5 Experiments and Results

**Dataset details**   Our method requires 2 images captured with different apertures (but same viewpoint) to integrate defocus cues. Standard monocular depth datasets [49, 35] typically capture in-focus images at a single aperture per scene, making them unsuitable for evaluating our method. While one could simulate defocused images with eq. (7) and RGBD data, this does not capture the model mismatch (occlusion, diffraction) in the physical image formation process. Therefore, to fairly evaluate our method, we construct a hardware capture setup, shown in fig. 2, comprising an Intel RealSense depth camera rigidly mounted to a DSLR camera (Canon EOS 5D Mark II). We use this system to collect a custom real-world dataset of 7 unique scenes, evaluating our method against learning-based MMDE baselines. We choose scenes with diverse subjects and depth profiles, placed within the operating depth range of RealSense (0.3–3.8m) to ensure accurate ground truth depth. Note that the CoC changes negligibly with depth beyond these distances, making defocus cues unreliable. For each scene, we capture images at 6 different apertures, $f/4, f/8, f/11, f/13, f/16$, and $f/22$, with the latter serving as the AIF image $\mathbf{x}$. The blurred image, $\mathbf{x}_b$, is selected from the lower F-stop images (see fig. 5 for comparison of F-stop setting on depth map quality). As described in section 4, we try to maintain a similar ratio of the exposure time and the camera aperture area for all the measurements taken for a scene. The lens focal length ($f$) and F-stop ($N$) are provided by the camera EXIF data, and the focus distance $F$ is read manually from the lens's analog focus scale[4].

---

[4]Alternatively, a lens with focus motor encoding would allow this to be known from EXIF

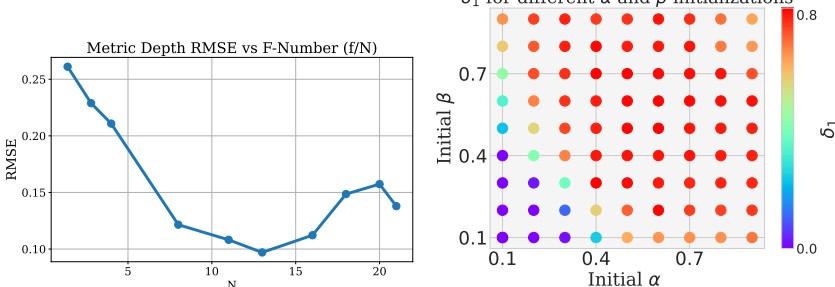

Figure 5: **Analyzing the effect of different aperture sizes and initializations.** *Left:* We use our forward model to simulate blurred images of a scene from the NYU-v2 dataset. We observe minimum depth error at $N = 13$, with errors increasing at more extreme aperture values. *Right:* We plot $\delta_1$ at the end of optimization for various $\alpha, \beta$ initializations normalized between 0-1 $\left( \alpha \rightarrow \frac{\alpha}{s_{\max}}, \beta \rightarrow \frac{\beta}{s_{\min}} \right)$, in the figure for visualization. While the performance degrades for small values of $\alpha, \beta$ (bottom left), it is relatively stable for a broad range of initializations.

Please see supplement.B for camera parameters and other dataset details, and supplement.H for more discussion on ground truth depth map quality.

**Evaluation Metrics**    We evaluate the predicted metric depth, $\hat{\mathbf{d}}^{\mathrm{m}}$, from our method against the RealSense ground truth depth $\mathbf{d}$ (with overloaded notation), using the metrics in [5, 39]. Specifically, we compute absolute relative error (REL) $= \frac{1}{M} \sum_{i=1}^{M} \frac{|\mathbf{d}_i - \hat{\mathbf{d}}_i^{\mathrm{m}}|}{\mathbf{d}_i}$, root mean squared error (RMSE) $= \sqrt{\frac{1}{M} \sum_{i=1}^{M} |\mathbf{d}_i - \hat{\mathbf{d}}_i^{\mathrm{m}}|^2}$, average log error (log10) $= \frac{1}{M} \sum_{i=1}^{M} |\log_{10} \mathbf{d}_i - \log_{10} \hat{\mathbf{d}}_i^{\mathrm{m}}|$, and accuracy thresholds $\delta_n =$ fraction of pixels where $\max \left( \frac{\mathbf{d}_i}{\hat{\mathbf{d}}_i^{\mathrm{m}}}, \frac{\hat{\mathbf{d}}_i^{\mathrm{m}}}{\mathbf{d}_i} \right) < (1.25)^n$ for $n = 1, 2, 3$. $M$ denotes the number of pixels, and $\mathbf{d}_i, \hat{\mathbf{d}}_i^{\mathrm{m}}$ denote the RealSense and predicted depths at pixel $i$, respectively. We also report point-cloud-based metrics, i.e., Chamfer Distance (CD) and the aggregated F1 score (FA) as defined in [39]. We align the ground truth and predicted depth map, similar to [19], and compute the metrics for pixels with non-zero values across the aligned depth maps. See supplement.C for more details on the depth map alignment/calibration procedure.

**Quantitative and qualitative results**    We evaluate our method against some of the recently popular MMDE methods UniDepth [39], Metric3D [74], and MLPro [7] on our collected dataset (table 2) and the NYUv2 [35] test set. Both Metric3D and UniDepth are provided with the required camera parameters as input. We outperform existing methods qualitatively (fig. 4) and quantitatively (table 2) on all the evaluation metrics, averaged over all 7 scenes in our collected dataset. Please see supplement.D for per-scene quantitative metrics on our dataset and supplement.H for depth error visualizations. On the NYUv2 test set, our method is on par with the MMDE baselines. See supplement.J for more details. For our dataset, the MMDE methods (MLPro, Metric3D) recover sharper details and are on-par with our method on some scenes (see fig. 4), but they lack consistency in their overall performance across all scenes. Our method achieves better consistency across varying scene conditions. Leveraging defocus cues enables our approach to recover the correct depth scale while also resolving relative depth errors in some cases (fig. 4 insets). We also evaluate our method (row 4 in table 2) using a Gaussian PSF [20] in the forward model. While previous work [20] uses the Gaussian PSF for training models for unsupervised depth recovery, we find that the model mismatch between the Gaussian PSF and the PSF of the real camera (fig. 2) leads to severe performance degradation in our case, compared to using the Disc PSF, which matches the real camera PSF better. This highlights the value of a physically consistent forward model, even with strong learned priors.

**Comparing with fine-tuning based methods**    Our method relies on optimizing noise-latents of a relative depth foundation model (RDFM) such as Marigold or Geowizard at test-time. Our approach (inference-time optimization) can be valuable in zero-shot settings, where fine-tuning-based methods would struggle due to the domain gap. We validate this by comparing our method with MMDE

| Method | RMSE ↓ | REL ↓ | log10 ↓ | $\delta_1$ ↑ | $\delta_2$ ↑ | $\delta_3$ ↑ | CD ↓ | FA ↑ |
|---|---|---|---|---|---|---|---|---|
| [71] FT Hypersim | 0.523 | 0.314 | 0.151 | 0.178 | 0.825 | 0.971 | 0.208 | 0.616 |
| [71] FT NYUv2 | 0.407 | 0.252 | 0.096 | 0.654 | 0.832 | 0.988 | 0.183 | 0.733 |
| Marigold FT NYUv2 | 0.475 | 0.231 | 0.098 | 0.634 | 0.878 | 0.963 | 0.175 | 0.808 |
| Ours, No FT | 0.273 | 0.125 | 0.052 | 0.879 | 0.975 | 0.991 | 0.103 | 0.870 |

Table 1: **Comparison with fine-tuning-based (FT) methods on our dataset.** Our method (row 4) outperforms (FT)-based methods across all metrics. NYUv2 (real, depth $\leq$ 10m) has a lesser domain gap to our dataset (real, depth $\leq$ 3.5m) than HyperSim (synthetic, depth $\leq$ 20m); as evident by the improvement in DepthAnythingv2 [71] when FT on NYUv2 (row 2) compared to HyperSim (row 1).

models obtained by fine-tuning RDFMs (DepthAnythingv2 [71], Marigold) on Metric depth datasets with a restricted depth range, as required by our method (section 4). We evaluate the following fine-tuning-based baselines on our collected dataset in table 1 – DepthAnythingv2 [71] fine-tuned on the Hypersim[42] dataset (row 1), NYU-v2 train set (row 2), and Marigold Fine-Tuned on NYUv2 (row 3). Our method (row 4, table 1) outperforms the fine-tuning-based baselines. This validates the utility of our inference-time method in zero-shot settings, where fine-tuning-based methods can struggle due to a domain gap. Please see supplement.K for details on fine-tuning these baselines and additional results on the NYUv2 test set. In addition to fine-tuning-based methods, we also outperform [56], which estimates metric depth from multi-aperture inputs via self-supervised learning with a differentiable forward model. See Supplement.L for more details on the comparison with [56].

**Extending our method beyond Marigold** Our method is plug-and-play and, in principle, works with any diffusion-based model like Marigold that exposes a differentiable mapping from its latent space to depth estimates. This enables test-time latent optimization, which allows for incorporating defocus cues without retraining. We demonstrate this by using Geowizard as the diffusion backbone [16] instead of Marigold. GeoWizard is a stable-diffusion-based monocular depth + normal predictor, trained on a more complex data distribution. Our method achieves similar performance with either of the backbones (Rows 5,6 in table 2), but shows minor patch-level artifacts with GeoWizard, likely due to stronger texture-depth coupling. See Supplement.I for visualizations and hyperparameter details.

| Method | RMSE ↓ | REL ↓ | log10 ↓ | $\delta_1$ ↑ | $\delta_2$ ↑ | $\delta_3$ ↑ | CD ↓ | FA ↑ |
|---|---|---|---|---|---|---|---|---|
| MLPro | 0.468 | 0.246 | 0.105 | 0.597 | 0.821 | 0.990 | 0.205 | 0.696 |
| UniDepth | 0.574 | 0.358 | 0.152 | 0.263 | 0.757 | 0.902 | 0.260 | 0.612 |
| Metric3D | 0.349 | 0.195 | 0.087 | 0.611 | 0.958 | 0.983 | 0.135 | 0.814 |
| Ours - Gaussian | 0.528 | 0.279 | 0.142 | 0.422 | 0.695 | 0.928 | 0.241 | 0.652 |
| **Ours - Disc** | **0.273** | **0.125** | **0.052** | **0.879** | **0.975** | **0.991** | **0.103** | **0.870** |
| Ours (Disc) with GeoWizard | 0.291 | 0.137 | 0.061 | 0.824 | 0.966 | 0.990 | 0.105 | 0.874 |

Table 2: **Comparison with learning based MMDE methods on our dataset.** Our method with the Disc PSF outperforms all the MMDE baselines averaged over all scenes in our dataset. The disc PSF, being more consistent with the real camera PSF, outperforms the Gaussian PSF. UniDepth, Metric3D, and our method are provided with camera intrinsics parameters during inference.

## 5.1 Ablation Studies

**Only single blurred image as input:** In our method, Marigold takes the all-in-focus (AIF) image (F/22) as input to predict relative depth, which is combined with learnable scale–offset parameters to synthesize a blurred image. The synthesized image is compared with a captured blurred image (F/8) that provides metric depth cues [52]. To assess the contribution of the AIF, we remove the AIF and instead provide a single moderately blurred image (F/16) as input to both Marigold and the loss function (eq. (11)). This ablation also evaluates whether the diffusion prior alone is strong enough to operate with a single modestly blurred image. Removing the AIF input leads to a large RMSE increase (1.36 vs. 0.346) and visible artifacts (fig. 6), indicating that while the diffusion prior captures coarse depth cues, the AIF input is essential for accurate metric depth estimation.

**Sensitivity to $\alpha, \beta$ initialization:** Gradient-based methods are known to be susceptible to local minima for non-convex optimizations. We thus evaluate the sensitivity of our method to the initialization for $\alpha, \beta$. We run the optimization in (eq. (11)) with a fixed $\mathbf{z}_T^{(\mathbf{d})}$ while grid-searching over initial

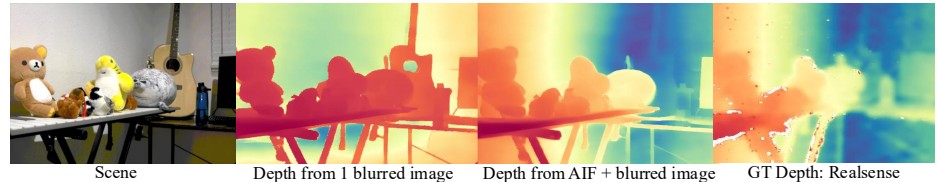

| Scene | Depth from 1 blurred image | Depth from AIF + blurred image | GT Depth: Realsense |

Figure 6: **Degradation in depth quality on using only a single blurred image.** For the TOYS scene, we observe that using a single blurred image as input results in severely inaccurate relative depth (middle), with all toys, guitar, and the monitor at similar relative depths. Our proposed method (right) recovers the depth ordering between the objects more accurately.

values of $\alpha$ and $\beta$. In fig. 5 (right), we observe that performance (measured by $\delta_1$) drops for small initialization values but remains stable across a broad range around $\alpha = 0.5, \beta = 0.5$, supporting the robustness of our chosen initialization. We also ablate on sensitivity to initial $\mathbf{z}_T^{(\mathbf{d})}$ in supplement.E.

**Improvements in relative depth from defocus cues** We quantitatively evaluate the effect of defocus cues in our method on improving the relative depth quality. Optimizing only $\alpha, \beta$ while holding $\mathbf{z}_T^{(\mathbf{d})}$ constant leads to a performance drop (table 3). This highlights the role of defocus cues in refining the relative depth initially predicted by Marigold. Please see supplement.F for visualizations.

| Method | RMSE $\downarrow$ | REL $\downarrow$ | log10 $\downarrow$ | $\delta_1$ $\uparrow$ | $\delta_2$ $\uparrow$ | $\delta_3$ $\uparrow$ |
|---|---|---|---|---|---|---|
| Ours $\alpha, \beta$ opt | 0.297 | 0.156 | 0.069 | 0.743 | 0.957 | 0.99 |
| **Ours** | **0.273** | **0.125** | **0.052** | **0.879** | **0.975** | **0.991** |

Table 3: **Relative depth quality** Optimizing the noise latent along with the affine parameters (ours) performs better than optimizing only the affine parameters ($\alpha, \beta$ opt).

**Different aperture sizes** We analyze how the aperture (F-stop) used for capturing the blurred image affects our performance. To do this, we use a scene from the NYU-v2 [35], an indoor RGBD dataset with high-quality ground truth depth annotations. This synthetic setup allows evaluating large F-stops that cannot be captured with our camera, while isolating aperture size from forward model mismatches in a real setup. Using the ground truth depth and our forward model (eq. (7)), we simulate the blurred images $\mathbf{x}_b$ at varying F-stops ($N$ values) and compute the error metrics between the ground truth and our predicted depth. We observe in fig. 5 that the performance (measured in RMSE) degrades for extreme aperture sizes. This is expected, as extreme blur (high or low) makes the inverse problem ill-posed, and an optimal blur level is key for accurate depth recovery. While $N = 13$ appears to be optimal in simulation (ignoring model mismatch), it underperforms $N = 8$ on average for real scenes, likely due to low contrast and insufficient blur cues in some of the scenes (STAIRS, PLANE). Please see supplement.G for results across all apertures captured in the real dataset.

## 6 Limitations and Future Work

While our method outperforms data-driven MMDE baselines, it remains significantly slower at inference time. Our method is best suited to scenes with a small depth range for which defocus blur offers high depth sensitivity. We observe that if the initial Marigold prediction is severely incorrect in some regions (visualized in supplement.F), the optimization may not always be able to fully correct them (ours for SHOERACK, THORDOG in fig. 4). A possible extension of our method is to jointly estimate the AIF and depth map from a single blurred input, as previously explored in non-zero-shot approaches [19, 1]. Our framework can broaden the utility of pre-trained depth priors to scientific applications involving depth-dependent imaging processes such as hyperspectral imaging [27], endoscopy [31], and microscopy [72]. While we avoid discretizing the depth map [23], our forward model loses accuracy at occlusion boundaries. We envision further improvements through better PSF engineering (coded aperture masks) and more accurate forward modeling of defocus blur. Another promising direction is to adapt feed-forward methods such as [71] for handling multi-aperture inputs through test-time adaptation and fine-tuning [73]. Large-scale multi-aperture datasets could make such methods practical for these settings.

**Acknowledgements:** We thank Alankar Kotwal and Bhargav Ghanekar for helpful discussions, Namrata Mantri for helping collect the real dataset, and Agastya Kalra for proofreading the draft.

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
