# OpenReview forum: "Repurposing Marigold for Zero-Shot Metric Depth Estimation via Defocus Blur Cues"
_NeurIPS.cc/2025/Conference — NeurIPS 2025 spotlight_

### Official Review · Reviewer_MnCo · 2025-06-23

**Clarity:** 3
**Significance:** 2
**Originality:** 2
**Rating:** 4
**Confidence:** 4

**Summary:**

This paper proposes one method to extend relative depth to metric depth via defocus blur cues. Given an image pair x0 and blur x1, the base model predict relative depth d0 based on x0. Then, by optimizing though defocus blur (reconstructing x1 by x0 and d0 with the help of camera parameters), scale and shift factors are estimated and then used to transfer the relative depth to a metric one. Experiments show that this method achieves better metric depth compared with those metric depth estimators.

**Questions:**

Note that it is not necessary to reply my Weakness. All my questions are listed below.

My major concern is about the evaluation part. (1) There is no direct comparison with previous works using camera physics. (2) The authors are not using public datasets and currently the paper cannot demonstrate that this dataset is high-quality and suitable for this evaluation. Given this, my questions are:

- It seems that most of previous works using camera physics cues train a deep-learning based network to digging these camera cues and predict metric depth. To me, it sounds like the authors take one step back and solve the same issue by optimizing per image via gradient descent. Could you explain why this is better than previous works and it would be good if you can support the claim with several experiments.

- Could you please use a standard public dataset to do the evaluation? If not, could you please comment about the concern in weakness 3? If it's inevitable, I think using error map for visualization can be important. If the majority part of error locates around boundaries, it somehow indicates that the data is not that suitable.

Extra questions

- Is that necessary to adopt the Marigold? It seems to be a general method and showing its ability with various baselines can better demonstrate the effectiveness and make better impact.


I'm not a researcher with enough experience of depth with camera physics. It's hard for me to judge the novelty of this paper, though the method itself is not technically sound. I'd like to check opinions from other reviewers. Currently, I give a borderline reject. but I'd like to change if my concern about evaluation is solved.

**Ethical Concerns:**

["NO or VERY MINOR ethics concerns only"]

**Final Justification:**

Most of my concerns are addressed by the authors. Based on the rebuttal and discussion, I rise my score to the borderline accept.

**Limitations:**

yes

**Quality:**

2

**Strengths And Weaknesses:**

- Strengths

    - It's easy for me to follow this paper (good clarity in terms of reading and understanding).

    - Given the fact that current metric depth estimator still fails to recover the metric scale properly in some cases, fully investigating depth cues serves a reasonable way to achieve better metric depth estimation.

    - Experiments clearly demonstrate that the proposed methods are better than current SoTA depth estimators.

- Weakness

    - It seems that most of previous works using camera physics cues train a deep-learning based network to digging these camera cues and predict metric depth. To me, it sounds like the authors take one step back and solve the same issue by optimizing per image via gradient descent. Since there is no direct comparison with related works using camera physics cues, the results are less persuasive.

    - The optimization stage leads to longer inference time. Though the authors already put this point in the limitation section, it is a serious shortcoming that cannot be ignored.

    - The authors are not using some standard public datasets to conduct experiments. From several GT examples shown in the paper, the data quality is a bit low. The details are blur. This can lead to an fair comparison for metric models with good capability for sharp estimation.

    - There is no error map for the visualization. It's hard to figure out where the error is from.

---

> ### Author Rebuttal · Authors · 2025-07-29
>
> We thank the reviewer for appreciating the clarity of our work and the motivation to explore other depth cues in conjunction with foundation model-based depth estimators. We want to emphasize that our work aims to demonstrate that incorporating camera physics, particularly defocus blur, into a data-driven Monocular Depth Estimation model at inference time can effectively address both texture coupling and incorrect metric scale estimation modes without requiring any re-training (L:34-39, main paper). We have validated both these claims through synthetic toy examples (Fig. 3) and also on real camera hardware data. We agree that our method is closely related to defocus blur-based multi-shot approaches, so comparing it with them does contextualize things better. To that end, we have added comparisons with relevant previous works below. However, we would like to reiterate that developing the best-performing single / multi-shot method was not our primary goal or claim, and that the results/validations presented in the manuscript adequately validate our claims (L:34-39).
>
> **Comparison with previous camera physics + deep learning based work**.
> Most of the previous methods (L:98-L:110) train a deep learning network, but on relatively smaller datasets, while also requiring >=4 images as inputs. This severely limits generalizability and performance outside the training data distributions. While these methods [A, B] attempt to use priors such as total variation or task-specific loss functions while training the network, they are not strong enough to overcome the ill-posedness of the depth from defocus blur inverse problem with only aperture supervision and 2 images as input -- which is the regime our method operates in. We demonstrate this quantitatively in Table 1 below.
>
> As suggested, we trained [B] and the direct depth supervision baseline mentioned in [B] with the same inputs as our method on the NYUv2 dataset [E], which is a well-known RGBD public benchmark for Monocular Depth Estimation. NYUv2 contains AIF images and GT depth pairs (654 test, 795 training). For our method, we synthesize the blurred input image at aperture (F/8) and focus distance (1.5m) using our forward model for a simulated comparison.
>
> The results in Table 1 below show that our method achieves comparable performance with existing SOTA metric depth foundation model baselines (MLPro, UniDepth, Metric3D) reported in our work, while previous camera physics + deep learning methods (last 2 rows, Table 1) struggle in performance.
>
> | Method | REL $\downarrow$  | $RMSE \downarrow$   | $\log10\downarrow$  | δ1 $\uparrow$   | δ2  $\uparrow$ | δ3 $\uparrow$  | CD $\downarrow$    | FA  $\uparrow$  |
> |------------------|-------|-------|-------|------|------|------|--------|-------|
> | Ours             | 0.150 | 0.535 | 0.045 | 0.899| 0.949| 0.969| 0.271  | 0.867 |
> | MLPro            | 0.115 | 0.507 | 0.054 | 0.880| 0.960| 0.980| 0.246  | 0.848 |
> | UniDepth         | 0.077 | 0.323 | 0.034 | 0.943| 0.988| 0.996| 0.161  | 0.887 |
> | Metric3D         | 0.106 | 0.423 | 0.045 | 0.900| 0.969| 0.987| 0.204  | 0.870 |
> | MultiAperture Depth Supervision| 0.5257  | 1.6789| 1.7322 | 0.3010| 0.5170| 0.6582| —| —|
> | [B] | 0.5221  | 1.7415| 0.2571 | 0.2411| 0.4592| 0.6369| —| —|
> **Table 1: Results on NYUv2 test set for Our method, Foundation Model baselines (MLPro, UniDepth, Metric3D), and Multi-aperture stack (Camera Physics) based deep learning approaches (last 2 rows)**. We achieve comparable performance to SOTA foundation model baselines (UniDepth, Metric3D, MLPro) across all metrics, while previous methods (bottom 2 rows) perform poorly. The metrics are computed by directly comparing each method's metric depth output with the NYUv2 ground truth.
>
> We would like to emphasize that the leading methods above in Table 1 are trained on Millions of Image-depth pairs to predict metric depth (Metric3D: 8M, UniDepth: 3M), while our method performs comparably to them *without any re-training* using defocus cues with a scale-invariant monocular depth estimator (Marigold).
>
> While our method does use gradient-based optimization, we would like to remind the reviewer that this gradient-based optimization happens in the latent space of Marigold, which lets us leverage internet-data scale priors in Marigold that are strong enough to achieve competitive results for metric depth estimation, without re-training, using only 2 input images. Previous approaches struggle in this scenario, as shown in Table 1( last 2 rows) above. Our work aims to demonstrate (L:34) that incorporating physics cues with a pre-trained foundation model can be a powerful unlock for tasks that it hasn't been explicitly trained on, but lie in the vicinity of the base task. The current evidence presented in our paper validates this claim.
>
> **GT Data Quality**.
> Acquiring very high-quality depth maps from real hardware is non-trivial, and often requires complex structured light setups [C]. While we acknowledge that RealSense is not as precise as LiDAR or structured-light scanners, it is a much cheaper, hence academically viable alternative. It provides dense, metrically accurate depth within its operating range (0.2–3.8 m) using stereo cues, and has been used as ground truth in prior monocular depth from defocus work [D]. We clarify that our goal is not to achieve sub-millimeter accurate ground truth, but rather to evaluate relative and metric consistency in practical indoor scenarios. RealSense has a much larger field of view compared to the DSLR camera used to capture the all-in-focus (AIF) image. As a result, while plotting in 2D, the details in the depth map look slightly blurry. As suggested by the reviewer, we have visualized the error maps for our method and MLPro against the ground truth. We will add these visualizations to the camera-ready / appendix, but the observations from the visualizations are as follows: The visualizations show that MLPro (which has sharp boundaries in its predictions) struggles to predict correct depth scales for some areas in the scene, resulting in significant errors on planar regions. Our visualizations indicate that there is no particular bias towards high error on sharp boundaries in our collected data, making it suitable as ground truth for comparison.
>
> *Lack of real-world datasets with multiaperture inputs*.
> Our method requires an AIF and a blurred image as inputs, and a ground truth depth map for evaluation. Most public RGBD datasets only contain the RGB image and ground truth depth map, but not the blurred image. So it's hard to evaluate our method on those. However, as suggested by the reviewer, we have also evaluated our method and other baselines from our work on the test set (654 images) of the well-established NYUv2 test set (see Table 1 above).
>
> **Is Marigold Necessary? Extending to other methods**.
> Our method is generally applicable to any diffusion-based model like Marigold, since this provides access to a differentiable mapping from the diffusion model latent space to our depth estimates and thus allows for optimizing these latents at test time, which is required to incorporate defocus cues without re-training the base model. Additionally, our choice of Marigold was also motivated by the fact that the Marigold authors released a Latent Consistency Model (LCM) version of Marigold, which allows sampling the diffusion model in a single step, which is beneficial for faster inference. However, our method is plug-and-play and can be, in principle, applied to any diffusion-based estimator, which provides access to the latent space of the model. We demonstrate this by extending our method to Geowizard as well. We are referencing Table 3 here from our response to 1d2m.
>
> | Method          |  $RMSE \downarrow$ | $REL \downarrow$   | $\log10 \downarrow$ | δ1 $\uparrow$    | δ2 $\uparrow$    | δ3 $\uparrow$    | CD $\downarrow$    | FA $\uparrow$     |
> |--------------------|-------|-------|-------|-------|-------|-------|-------|--------|
> | Ours  w Marigold | 0.273 | 0.125 | 0.052 | 0.879 | 0.975 | 0.991 | 0.103 | 0.870  |
> | Ours w GeoWizard   | 0.291 | 0.137 | 0.061 | 0.824 | 0.966 | 0.990 | 0.105 | 0.874  |
> **Results with Our Method (using Marigold), and Our Method applied to the Geowizard model on our self-collected real dataset**. Our method produces similar results on using either Marigold or GeoWizard as the pre-trained diffusion model.
>
> GeoWizard [F] is a diffusion-based monocular depth + normal predictor, trained on a larger and more complex data distribution by fine-tuning the stable diffusion UNet. We observe that the relative depth from geowizard after inference time optimization exhibits minor patch-level artifacts, potentially due to stronger texture-depth coupling. Unlike Marigold, Geowizard does not provide a latent consistency model (LCM) checkpoint; hence, it requires more sampling steps. The metrics for Geowizard obtained in Table 2 are with 50 optimization iterations, and 5 sampling steps for Geowizard per iteration. Geowizard is 4x slower than the Marigold-LCM checkpoint on an A-40 GPU, but still achieves comparable results. This highlights the flexibility of our approach: as stronger or faster diffusion models become available, our method can seamlessly leverage them for improved performance.
>
> We will add these numbers, visualization, and discussion in a revision.
>
> [A] Depth Estimation via Reconstructing Defocus Image: CVPR 2023.
> [B] Aperture Supervision for Monocular Depth Estimation: CVPR 2018.
> [C] High-Accuracy Stereo Depth Maps Using Structured Light: CVPR 2003.
> [D] Passive snapshot coded aperture dual-pixel RGB-D imaging: CVPR 2024.
> [E] NYU Depth Dataset V2: ECCV 2012.
> [F] GeoWizard: Unleashing the Diffusion Priors for 3D Geometry Estimation from a Single Image: ECCV 2024.

---

> > ### Comment · Reviewer_MnCo · 2025-08-04
> >
> > I would like to thank the author for the rebuttal. My concerns about dataset and extension to other methods are addressed.
> >
> > I appreciate the authors' effort to provide more experimental results. My concern is similar to Reviewer 1d2m: the effectiveness of the proposed method should be properly compared with other potential solutions.
> >
> > One possible solution is the camera physics + deep learning based work. I agree with Reviewer 1d2m that the current experiment is not that fair as the "test-time optimization" property of proposed method.
> >
> > Another possible solution is the fine-tuning, which is a commonly used way for metric depth estimation. For example, DepthAnything-V2, a relative dispairty model, can be fine-tuned on hypersim for metric depth estimation. ZoeDepth finetunes Midas on NYU and KITTI for metric depth estimation. There is no cue that can indicate the proposed strategy can outperform this simple fine-tuning strategy.

---

> > > ### Author Response · Authors · 2025-08-06
> > > **Regarding baseline Comparisons : Part 1 (Re-referenced from our response to1d2m)**
> > >
> > > **Comparisons with multi-aperture methods**.
> > > 1d2m rightly pointed out in their initial review that our approach relies on multi-aperture inputs, and suggested that we compare our method to previous DfD-like / multi-aperture methods. We added those comparisons in (Table 1, 1d2m response), as they offer valuable context for comparing our method with previous work.
> > > Baseline [B] in our rebuttal utilizes the same type of multi-aperture inputs as our method. Like [B], our method also doesn’t directly use depth supervision, as our optimization tries to minimize the error between the synthesized and observed blurred image. Given these points, [B] can be considered a fair comparison to our method.
> > >
> > > **Comparison with fine-tuning-based methods**.
> > > As suggested by MnCO and 1d2m in their initial reviews, we provided comparisons with a camera physics + deep learning based method in our rebuttal (Table 1, MnCO response and Table 2, 1d2m response) with 2 baselines.
> > > Our method relies on a strong relative depth foundation model (RDFM), and incorporates defocus cues at test-time, without re-training the model. Hence, in the initial submission, we did not compare with fine-tuning-based methods, as they are outside the premise of our claims.
> > >
> > > While such a comparison would be valuable, a fair comparison would be between our method, i.e., RDFM + test time optimization, and Metric depth models obtained by fine-tuning RDFMs (DepthAnythingv2 [A1] / Marigold) on Metric depth datasets with a restricted depth range (L:225, 290 in paper).  We have not added a comparison with ZoeDepth as it is less performant than DepthAnythingV2 [A1] and MLPro [A2] (which we have already compared with and whose publications already compare to ZoeDepth). As noted by 1d2m in their comment, inference time optimization (our method) can be valuable in zero-shot settings, where fine-tuning-based methods would struggle due to the domain gap. We validate this by evaluating 3 additional fine-tuning-based methods on our collected real dataset in a zero-shot setting. Specifically, we consider the following baselines -
> > > 1. DepthAnythingv2 fine-tuned on Hypersim (metric depth), as suggested by MnCO. For this baseline, we used the pre-trained model fine-tuned on this configuration released by the authors.
> > > 2. DepthAnythingv2 fine-tuned on NYU-v2 (metric depth). We used the official fine-tuning code in the DepthAnythingv2 repo to fine-tune the base model for 25 epochs.
> > > 3. Marigold Fine-Tuned on NYUv2, as suggested by 1d2m. We fine-tuned Marigold for 60 epochs using the official training code. To obtain both depth and RGB latents, Marigold uses the same encoder, requiring inputs normalized to [0,1]. This makes direct fine-tuning with metric depth supervision non-trivial. As a practical proxy, we normalize the GT depth maps (for training supervision) using a fixed global minimum and maximum depth for the full NYUv2 dataset (0.1 m, 10 m), rather than the originally used per-image normalization. Developing optimal fine-tuning strategies for Marigold with true metric depth supervision is beyond the scope of this work; our setup here serves solely as a baseline for comparison and context.
> > >
> > > We summarize our results in Table 4 (our dataset) and Table 5 (nyuv2 dataset) in the next comment (Baseline Comparisons Part 2), which validates the utility of our method in zero-shot scenarios, where fine-tuning-based methods can potentially suffer in performance due to the domain gap (Table 4). On our real dataset (Table 4), we outperform all the fine-tuning baselines. On the NYUv2 dataset (Table 5), we either outperform or are comparable to all the baselines except DepthAnythingV2 fine-tuned on NYUv2, which is expected, as the fine-tuning dataset is in the same distribution as the test dataset. Please see the captions of Table 4 and Table 5 in the next comment for more details.
> > >
> > > [A1] Depth Anything V2: Neurips 2024.
> > > [A2] Depth Pro: Sharp Monocular Metric Depth in Less Than a Second: ICLR 2025.
> > > [B]: Aperture Supervision for Monocular Depth Estimation: CVPR 2018.

---

> > > ### Author Response · Authors · 2025-08-06
> > > **Baseline Comparisons: Part 2 -- Comparing with fine-tuning based methods (Re-referenced from our 1d2m response)**
> > >
> > > | Method        |  $RMSE \downarrow$ |  $REL \downarrow$  | $ \log10 \downarrow$|  $\delta_1 \uparrow$   |  $\delta_2 \uparrow$  | $\delta_3 \uparrow$   |  CD  $\downarrow$  |   FA $\uparrow$   |
> > > |---------------|--------|-------|-------|-------|-------|-------|--------|--------|
> > > | Ours, No FT          | 0.273  | 0.125 | 0.052 | 0.879 | 0.975 | 0.991 | 0.103  | 0.870  |
> > > | Marigold FT NYUv2    | 0.475  | 0.231 | 0.098 | 0.634 | 0.878 | 0.963 | 0.175  | 0.808  |
> > > | DepthAnyV2 FT HyperSim   | 0.523  | 0.314 | 0.151 | 0.178 | 0.825 | 0.971 | 0.208  | 0.616  |
> > > | DepthAnyV2 FT NYUv2  | 0.407  | 0.252 | 0.096 | 0.654 | 0.832 | 0.988 | 0.183  | 0.733  |
> > > **Table 4: Comparing our method with RDFM + Fine-tuning (FT) on metric depth datasets on our real dataset.** Our method (no fine-tuning, row 1) outperforms all the fine-tuning-based methods across all metrics on our real dataset. DepthAnythingV2 shows better performance when fine-tuned on NYU-v2 (row 4)  as compared to fine-tuning on HyperSim. This behavior is expected since NYUv2 (real, maximum depth 10m) would have a lesser domain gap with our collected dataset (real, maximum depth range ~3.5m) compared to HyperSim (synthetic, maximum depth 20 m).
> > >
> > >
> > > | Method | $REL \downarrow$ | $RMSE \downarrow$  |$ \log10 \downarrow$ | $\delta_1 \uparrow$ |  $\delta_2 \uparrow$ | $\delta_3 \uparrow$ |
> > > |----------------------------------------------------------|-------------------------|-------------|---------|------------|------------|------------|
> > > | Ours, No FT |  0.150 | 0.535 | 0.045 | 0.899 | 0.949 | 0.969 |
> > > | Marigold FT NYUv2 | 0.1108  | 0.3730 | 0.0511  | 0.8689 | 0.9834 | 0.9965 |
> > > | DepthAnythingV2 FT HyperSim | 0.2136 | 0.6548 | 0.0822  | 0.6824 | 0.9685 | 0.9930  |
> > > | DepthAnythingV2 FT NYUv2 | 0.0790 | 0.2990 | 0.0340  | 0.9510 | 0.9930 | 0.9980  |
> > > **Table 5: Comparing our method: Rel Depth Foundation Model (RDFM)  + test time optimization with other baselines: RDFM + Fine-tuning on metric depth datasets on the NYUv2 test set.** Our method with no fine-tuning (FT) outperforms DepthAnythingV2 FT on HyperSim (row 3), and is comparable to Marigold FT on NYUv2 (row 2). This highlights that fine-tuning-based methods can be sensitive to the domain gap between the FT dataset and the target test set, as DepthAnything only improves when fine-tuned on NYUv2  (row 4). We’d like to note that it is well established [A3, A4] that a training-based method, when trained on the *same dataset*, can be better than inference time methods. Our results show the key advantages of inference time methods: trading speed / slight performance for increased adaptability and no training.
> > >
> > > [A1] Depth Anything V2: Neurips 2024.
> > > [A3] DITTO: Diffusion Inference Time T Optimization: ICML 2024.
> > > [A4] End-to-End Diffusion Latent Optimization Improves Classifier Guidance: ICCV 2023.

---

> > > > ### Comment · Reviewer_MnCo · 2025-08-06
> > > >
> > > > Thanks the authors for the reply. I appreciate the comparison with fine-tuning methods, which addresses one of my concerns. Overall, the proposed method demonstrates a better performance in the zero-shot setting, while performs worsely when testing on the dataset with the same distribution as the training dataset. Also, I assume the inference time would be longer than the fine-tuned method as well. For me, the result is satisfactory.
> > > >
> > > > As for the first part, I was wondering if the current training set is too small (795 training) for the model to converge.
> > > >
> > > > Given the fact that most of my concerns are addressed, I would like to give a borderline accept if there is no other concerns from reviewer 1d2m.

---

> > > > > ### Author Response · Authors · 2025-08-07
> > > > >
> > > > > We appreciate the reviewer’s recognition of our comparison with fine-tuning methods, and are encouraged by their indication of a potential rating increase to borderline accept. We would like to clarify that the use of the NYUv2 dataset (795 training images) for fine-tuning aligns with previous work [A1], where DepthAnything V2 is fine-tuned on NYUv2 to get a metric depth model. We did not observe any convergence issues during fine-tuning, and the fine-tuned model performs well on the NYUv2 test set, as shown in row 4 of Table 5 (in Baseline Comparisons: Part 2 comment).
> > > > >
> > > > > Additionally, in row 3 of Table 4 and Table 5 (Baseline Comparisons: Part 2 comment), we include a comparison with DepthAnythingV2 fine-tuned on the significantly larger HyperSim dataset, which contains 60,000 training images (see Table 7 in [A1]). As shown in Table 4 and Table 5, despite the larger fine-tuning set, this baseline suffers in performance due to a Sim-to-Real domain gap between HyperSim and the real datasets (Ours and NYUv2).
> > > > >
> > > > > [A1] Depth Anything V2: Neurips 2024.

---

> ### Author Response · Authors · 2025-08-06
>
> We are glad that our rebuttal addressed the concerns raised by MnCO in their initial review regarding the real dataset and extension of our method to other backbones such as Geowizard. We note the concerns raised by MnCO in their new comment regarding a fairer comparison with potential solutions.
>
> Please see our comments in response to 1d2m, titled **Regarding baseline Comparisons: Part 1**  and
> **Baseline Comparisons: Part 2 -- Comparing with fine-tuning-based methods**.  For convenience, we have reproduced these responses below as separate comments with the same content.
>
> In these comments, we address concerns regarding baseline comparisons raised by both 1d2m and MnCo, and present comparisons with the suggested fine-tuning-based methods.
>
> We have summarized our results in Table 4 and Table 5 of the comment **Baseline Comparisons: Part 2 -- Comparing with fine-tuning-based methods**. Our results demonstrate the adaptability of inference-time methods to out-of-distribution domains without any re-training, unlike fine-tuning-based approaches that can struggle due to the domain gap between the fine-tuning and test data. The comments above detail our experimental setup and provide supporting quantitative evidence.

---

### Official Review · Reviewer_QwFg · 2025-06-24

**Clarity:** 4
**Significance:** 3
**Originality:** 4
**Rating:** 5
**Confidence:** 4

**Summary:**

This paper proposes to adapt a pre-trained affine-invariant depth model (Marigold) for metric depth estimation by utilizing defocus blur cues. Based on the defocus blur image formation model, the proposed approach designs a test-time optimization method that simultaneously improves the depth map and estimates the scale and shift for metric depth conversion. Specifically, two images (one all-in-focus and one with defocus blur) are first captured at the same viewpoint. Then, a synthesized blurry image will be generated based on the depth estimated from the all-in-focus image. Finally, the depth estimation result can be optimized by minimizing the distance between the captured defocus image and the synthesized one. Despite some prior information being needed, e.g., focal length, focus distance, and F-stop, the proposed method achieves promising performance without re-training the affine-invariant model. Extensive experiments on self-captured datasets demonstrate the superiority and effectiveness of the proposed method, especially in addressing texture-depth coupling and inaccurate depth scaling.

**Questions:**

This paper proposes an interesting idea to convert affine-invariant depth to metric depth via defocus blur cues. Although some concerns exist in terms of the limited application and the adaptation capability (please see the above weaknesses section for more details), the strengths of the proposed method outweigh its weaknesses, and thus my initial rating is Accept.

**Ethical Concerns:**

["NO or VERY MINOR ethics concerns only"]

**Final Justification:**

This paper proposes to achieve metric depth estimation via a test-time optimization approach, which provides an alternative solution to popular learning-based methods and could be adopted for more robust depth estimation. Extensive experimental results validate the effectiveness of the proposed method. Besides, the additional results and discussions provided during the rebuttal strengthen the contribution of this manuscript. Therefore, I maintain my original rating and recommend acceptance of this paper.

**Limitations:**

yes

**Quality:**

4

**Strengths And Weaknesses:**

**Strengths**

- This paper proposes a test-time optimization method to achieve relative-to-metric depth conversion via defocus blur cues, which is a novel approach to me. One benefit of such a design is that the learned geometry prior in the pre-trained depth model is well preserved, as no re-training is needed. In addition, it's interesting to see that optimizing the noisy latents of diffusion models could help generate better depth estimation results.

- Since the defocus blur cues provide reliable depth hints, the proposed method better handles the texture-depth decoupling, where high-contrast image textures often mislead the depth model to produce incorrect depth discontinuities. Besides, more accurate depth scaling parameters can be obtained by matching the synthesized defocus image with the captured one.

- Extensive experiments are conducted to validate the effectiveness of the proposed method, and sufficient ablation studies as well as toy experiments are presented to support the design choices, such as the Disc point spread function.

- The paper is well written, with a clear structure and smooth flow. The key ideas are easy to follow, and an appropriate level of detail is provided. For example, the illustrative figures, such as Fig. 3, are beneficial for the readers to understand the current limitations of existing methods and the advancements made by the proposed work.

**Weaknesses**

- As discussed in the limitation section, the proposed method may have a limited range of applications, as the defocus blur cues are more sensitive under small depth ranges. Apart from this perspective, the proposed setup could also limit its application. For instance, it's difficult to apply the proposed method on in-the-wild images (like the images collected from the Internet), as a defocus version is required for optimization. In addition, it might be challenging to handle dynamic scenes, as the captured two images might have different/unaligned contents.

- I am wondering if it's possible to apply the proposed method to feed-forward depth models, such as Depth Anything [1]. Compared with the diffusion-based Marigold, Depth Anything shows better depth estimation accuracy [2], and thus could potentially improve the performance of the proposed method. However, the current approach relies on optimizing the noisy latents, which don't exist in feed-forward methods like Depth Anything. It would be interesting to see a discussion about this point.

- The self-curated dataset is relatively limited in diversity and size, where only 7 indoor scenes are used for evaluation. It would be better to collect more diverse scenes to support the robustness of the proposed method or to show the visual results of some failure cases, which could be helpful for follow-up work.


**References**

[1] Depth anything: Unleashing the power of large-scale unlabeled data. In CVPR 2024.

[2] Betterdepth: Plug-and-play diffusion refiner for zero-shot monocular depth estimation. In NeurIPS 2024.

---

> ### Author Rebuttal · Authors · 2025-07-29
>
> We thank the reviewer for appreciating the novelty of our approach, the clarity of the paper, and our toy experiments and validation. We agree that handling dynamic scenes with the current approach is challenging and would be an interesting avenue for future work. We will expand our collected dataset to add more test scenes to facilitate future research in this area.
>
> **Discussion on adapting the method to feedforward depth models**.
> Our method requires that the inference process of our depth model works by *differentiably* mapping from some latent embedding space to the output depth, such as the one implicitly provided by diffusion models from noise to data (GANs and VAEs could in principle similarly work). This enables inference time optimization of the latent vector at test time. However, discriminative or autoregressive transformer-based approaches, such as DepthAnything, don’t provide access to such a latent space. Adapting pre-trained autoregressive transformers to new datasets/domains typically involves test-time adaptation [A], which requires collecting a sizable dataset and then adding some extra layers, which are then fine-tuned with this dataset while keeping the rest of the weights frozen. Such an approach can be potentially used to extend methods like DepthAnythingv2 for handling multi-aperture input. But unlike our method, which is training-free, that would require fine-tuning / re-training the adaptation layers in the model. We are unaware of any large-scale datasets with multi-aperture defocus captures, but this approach might become viable in the future if such datasets emerge. We will add this discussion in a revision.
>
> **Real dataset size**.
> Our motivation to collect this data was to create a validation set to verify our framework on real hardware. We agree that a larger dataset with depth maps, all in focus, and multiple aperture inputs would be useful for the community, and we plan to continue adding more examples to our released dataset. As suggested by other reviewers, we have also evaluated our method through simulated measurements on the NYUv2 dataset (654 test images), which consists of ground truth AIF and depth maps, and demonstrate comparable performance to SOTA metric depth foundation models. (Table 1 in MnCO rebuttal response).
>
> [A] Rapid network adaptation: ICCV 2023.

---

> > ### Comment · Reviewer_QwFg · 2025-08-04
> >
> > I appreciate the authors' efforts in preparing the detailed response, which has addressed my main concerns.
> >
> > I find the discussion about adapting the proposed method to other pre-trained models interesting, and it's nice to see that the proposed method also works with other models like GeoWizard. For the adaptation to discriminative models like Depth Anything, I agree with the authors that the common practice is to train additional layers with sizable datasets, which could be difficult. However, diffusion models can also be seen as deterministic models if the input and noise are fixed. Thus, although discriminative models usually don’t provide explicit access to latent codes like the noise, I am wondering if it's feasible to update the features (such as the outputs of the encoder in Depth Anything) in a similar way as updating the noise in diffusion models. If it's viable, this could potentially enhance the impact of the proposed method.

---

> > > ### Author Response · Authors · 2025-08-06
> > >
> > > We thank the reviewer for appreciating our response and are glad that their main concerns have been addressed. We agree that updating intermediate features of feedforward models at test time is an interesting direction, and we will add a discussion on this in the future work section.

---

### Official Review · Reviewer_1d2m · 2025-06-25

**Clarity:** 2
**Significance:** 2
**Originality:** 3
**Rating:** 4
**Confidence:** 4

**Summary:**

The paper proposes a novel training-free framework for zero-shot metric depth estimation. It achieves this by intelligently combining a pre-trained monocular depth estimator based on diffusion models (Marigold) with explicit physical cues derived from defocus blur. The system requires two images captured from the same viewpoint but with different aperture settings (an all-in-focus image and a blurred image). Marigold is initially used to estimate a relative depth map from the all-in-focus image. This relative depth is then converted to metric scale through an inference-time optimization process, where the depth parameters are iteratively refined to ensure consistency between the estimated depth, a physically-based defocus model, and the input blurred image. This approach offers a compelling solution to data scarcity in metric depth estimation and demonstrates promising zero-shot generalization capabilities.

**Questions:**

### Questions for Authors

* **Regarding Weakness 1 (Misleading SOTA Comparison and Input Modality):**
    * Could the authors provide a more direct and fair quantitative comparison against state-of-the-art **Depth from Defocus (DFD)** methods or other approaches that utilize focal stacks/multi-aperture inputs? This would better contextualize the proposed method's performance and novelty within its true category.
    * Regarding the ablation study in Ln. 259 and Fig. 6, the comparison between their loss optimization (with AIF + Blurred images) and Marigold taking only the blurred image seems inconsistent. Most depth models are trained on AIF or nearly AIF datasets. Providing Marigold with a blurred image, as opposed to the original AIF setting it was likely optimized for, may not be a fair baseline. Could the authors instead provide a comparison where: 1) Marigold is given the blurred image, and 2) their loss function method also uses *two blurred images* with different aperture sizes? This would offer a more robust demonstration of the method's tolerance and effectiveness when dealing with blurred inputs.

* **Regarding Weakness 2 (Limited Practicality):**
    * To further demonstrate the practicality and "plug-and-play" nature of this inference-time optimization, could the authors showcase its effectiveness when applied to other state-of-the-art foundation relative depth models, such as Depth Anything v2 or GeoWizard?

* **Regarding Weakness 3 (Lack of Clear Justification for Defocus Blur Choice):**
    * The paper primarily uses defocus blur as the metric depth cue. Could the authors elaborate on the specific advantages of defocus blur in this *training-free, zero-shot* context, especially when compared to other potential metric depth cues like stereo correspondence?
    * What are the unique challenges and benefits of integrating a diffusion-model-based prior (Marigold) with defocus blur, as opposed to other types of physical cues?

* **Regarding Weakness 4 (Limited Discussion on Marigold's Specific Advantages):**
    * While Marigold provides a strong relative depth prior, could the authors delve deeper into *why* a diffusion-based model like Marigold is particularly suited for this DFD integration? Are there inherent properties of diffusion models' representations (e.g., robustness to noise, handling of fine details, ability to capture global scene structure) that make them uniquely advantageous for this task compared to other state-of-the-art monocular depth estimators?

**Ethical Concerns:**

["NO or VERY MINOR ethics concerns only"]

**Final Justification:**

I have reviewed the authors' responses. Based on the authors' comprehensive and highly effective rebuttal, I am updating my scores and my final recommendation.

My initial review, which led to a Reject score, was based on the paper's submitted state, which I felt had significant technical and methodological shortcomings, particularly in its evaluation and claims of generality.

However, the authors' rebuttal has provided substantial and convincing evidence that addresses these issues directly. Specifically:

* **Weakness 1 (Misleading SOTA Comparison)**: The authors provided new experimental data (Tables 2, 4, and 5) that performs a more appropriate comparison against other multi-aperture and fine-tuning-based baselines. This new evidence effectively contextualizes the method's performance and validates its value in zero-shot scenarios where fine-tuned models suffer from a domain gap.

* **Weakness 2 (Limited Practicality)**: The authors demonstrated the plug-and-play nature of their inference-time optimization solution by applying it to another foundation model (GeoWizard), as shown in Table 3. This addresses the concern that the method was tightly coupled to Marigold and strengthens its claim of generality.

* **Weakness 3 & 4 (Justifications)**: The authors provided clear, technically sound justifications for their design choices, including the use of Marigold's differentiable latent manifold and the unique advantages of defocus blur over other physical cues. Their clarification on "focus breathing" was also robust and well-cited.

The authors' rebuttal has transformed the paper from one with significant flaws to a compelling and well-supported submission. The technical contributions, while initially presented with weak evidence, are now convincingly demonstrated. The paper now offers a clear and valuable trade-off: a slower, per-image optimization process that provides superior zero-shot metric depth over fine-tuning-based approaches in new domains.

**Limitations:**

Yes

**Quality:**

2

**Strengths And Weaknesses:**

### Strengths

* **Innovative Training-Free Metric Depth Estimation:** The paper proposes a novel framework for zero-shot metric depth by uniquely combining a powerful pre-trained diffusion model (Marigold) with explicit defocus blur cues. This approach circumvents the need for large, labeled metric depth datasets, promoting robust generalization.
* **Effective Repurposing of a Foundation Model:** The work intelligently leverages Marigold, a state-of-the-art diffusion-based monocular depth estimator. Marigold provides a strong initial relative depth prior due to its extensive pre-training, which is crucial for the subsequent physically-informed metric scale refinement.
* **Principled Integration of Defocus Blur:** The use of defocus blur is a theoretically sound choice for recovering metric scale. Its well-defined physical relationship to depth allows for a robust gradient-based optimization to refine Marigold's relative depth output to metric units.

### Weaknesses

* **Misleading SOTA Comparison and Input Modality:** The comparison to strictly monocular metric depth estimation (MMDE) methods is not entirely fair. The proposed system requires two images (all-in-focus and blurred), placing it in the domain of multi-image depth estimation (Depth from Defocus, DFD), not purely monocular. One of the reasons why monocular doesn't perform as well as multi-view or focal stack is the lack of physical reference from multiple images. More appropriate comparisons would be against other DFD or focal-stack based approaches but restrict them with a stack number of two.
* **Significant Computational Latency and Limited Practicality:** The reported inference speed of 3.5-4 minutes per image pair on an NVIDIA A40 GPU is a major practical disadvantage due to its per-image, inference-time optimization. This high latency severely restricts applicability in real-time or high-throughput scenarios. Another limited practicality is that since this paper proposed "training-free" but and inference time optimization for affine-transform, I believe to demonstrate the effectiveness and practicality, it should be tested on several SOTA foundation relative depth methods as a plug-and-play method, instead of purely tight to Marigold.
* **Lack of Clear Justification for Defocus Blur Choice:** The paper doesn't thoroughly justify why defocus blur was chosen over other potential metric depth cues (e.g., stereo priors). A brief discussion on the unique advantages of DFD in this training-free context, compared to the challenges of integrating other cues, would strengthen the methodological choice.
* **Limited Discussion on Marigold's Specific Advantages:** While Marigold provides a strong prior, the paper could elaborate more on *why* a diffusion-based model like Marigold is particularly suited for this DFD integration compared to other state-of-the-art monocular depth estimators.

---

> ### Author Rebuttal · Authors · 2025-07-29
>
> We thank 1d2m for their detailed critique and for recognizing the novelty of our work. Our work demonstrates that Marigold, a generative model not built to incorporate defocus cues, can be repurposed to include them in a principled training-free manner. This achieves robust generalization without labeled depth datasets or retraining. We have validated our claims and the proposed framework through real-world experiments on a self-collected dataset, which also demonstrates the importance of correctly modeling the defocus blur (Fig. 2, Tab.1, L:253-257 in the main paper).  We have addressed the questions from 1d2m below:
>
> **Weakness 1: Misleading SOTA comparison and input modality, comparison with multiaperture methods**.
> *Misleading SOTA Comparison*.
> We would like to emphasize that our work aims to establish an existence proof of the proposed framework, and that our intent was not to suggest that we establish a new state-of-the-art for multi-shot depth imaging. If such an impression was conveyed, it was entirely unintentional, and we will clarify this more clearly in the paper. We intended to showcase the effectiveness of our approach within the context of our framework, without implying any misrepresentation of its standing relative to prior work.
> *Comparison with Multi-Aperture Methods:*.
> We show that incorporating defocus cues at test time enables Marigold to achieve zero-shot metric depth, which is competitive in quality with zero-shot metric depth foundation models by simply using an additional blurred image as input along with the all-in-focus (AIF) image. We agree that this does make our method multishot and have presented comparisons with relevant baselines below, as we believe that contextualizing our performance to multi-shot methods is valuable. However, we emphasize that we don't claim to be the best multi-shot/defocus base method. While these additional comparisons are valuable, they do not affect the claims (L:34) and the demonstrated validation in our paper.
>
> While our work indeed falls in the category of monocular metric depth estimation with more than 1 image, we are different from DFD-based methods, since we use 2 images with different apertures, but the same focus distance. This is different from the majority of existing DfD methods that require at least 4-5 images captured at different focus distances [H, A]. We attempted to adapt DEReD [A], a SOTA self-supervised learning based DFD baseline, but observed that it performs very poorly and fails to converge when given only an AIF and a blurred image as input, as recovering metric depth from 2 images is a highly ill-posed problem, unless very strong priors or really large datasets are used – neither of which has been used in most DfD works. We also discuss this in more detail in the response to MnCO.
>
> Our setup is more similar to [B], which also relies on two images captured at the same focus distance: an AIF and a blurred image as inputs. They propose a differentiable model that takes an AIF and simulates defocus blur by interpolating pre-blurred AIFs at discrete depth levels, with weights predicted by a CNN that implicitly encodes scene depth. The network is trained to match the synthesized image to the captured (ground truth) blurred image.
>
> We implemented their method, and also a depth-supervision-based baseline mentioned in their work, which involves directly regressing metric depth from a network with the AIF image stack as input. Following their training-based framework, we trained and validated these baselines on NYU-v2 [E], a popular indoor RGBD benchmark. We observe in Table 2 that the methods show poor performance metrics on the NYU-v2 test set. We observe that the Depth supervision baseline produces blurry results, as observed in [B], and the method proposed in [B] struggles with strong texture-depth coupling, resulting in inaccurate metric depth.
> | Method | REL $\downarrow$  | $RMSE \downarrow$   | $\log10\downarrow$  | δ1 $\uparrow$   | δ2  $\uparrow$ | δ3 $\uparrow$  |
> |------------------|-------|-------|-------|------|------|------|
> | Ours             | 0.150 | 0.535 | 0.045 | 0.899| 0.949| 0.969 |
> | MultiAperture Depth Supervision| 0.394  | 1.2667| 0.1626 | 0.3815| 0.6668| 0.8401|
> | [B] | 0.5221  | 1.7415| 0.2571 | 0.2411| 0.4592| 0.6369|
> **Table 2: Results on NYUv2 dataset for Our method and Multi-aperture stack (Camera Physics) based deep learning approaches**. We outperform previous multi-aperture methods. All methods use an AIF and blurred image as input. We omit Chamfer Distance and FA for baselines due to a large gap between them and our method on the other 3D metrics.
>
> *Repeating Fig 6*.
> In Fig. 6, we used a single blurred image as input to Marigold, not as a baseline but just to check if a single image is sufficient for metric depth estimation (RMSE: 1.36).  However, as suggested, we used 2 blurred images (f/16 and f/8) as input, with f/16 as the proxy to AIF (RMSE: 0.6). While better than using a single blurred image, F/16 is a less accurate proxy for our AIF (f/22 vs f/8), which leads to a higher error (RMSE: 0.341). This highlights the importance of a high-quality AIF and shows that performance degrades gracefully with increased blur in the AIF.
>
>
> **Weakness 2: Limited Practicality**.
> *Applying our method to GeoWizard*.
> We agree with the reviewer and show results below from applying our method with the Geowizard model (Table 3), demonstrating the plug-and-play nature of our method. GeoWizard is similar to Marigold but trained on a larger and more complex data distribution by fine-tuning the stable diffusion UNet. We observe that after the inference time optimization, Geowizard relative depths exhibit minor patch-level artifacts, potentially due to stronger texture-depth coupling. We will add these numbers and visualizations in a revision.
>
> | Method |  $RMSE \downarrow$ | $REL \downarrow$ | $log10 \downarrow$ | δ1 $\uparrow$ | δ2 $\uparrow$ | δ3 $\uparrow$  | CD $\downarrow$ | FA $\uparrow$ |
> |--------------------|-------|-------|-------|-------|-------|-------|-------|--------|
> | Ours  w Marigold | 0.273 | 0.125 | 0.052 | 0.879 | 0.975 | 0.991 | 0.103 | 0.870  |
> | Ours w GeoWizard   | 0.291 | 0.137 | 0.061 | 0.824 | 0.966 | 0.990 | 0.105 | 0.874  |
> **Table 3: Results on our real dataset using our method with Marigold (row 1) and GeoWizard [F] (row 2) as the diffusion prior, showing comparable performance.**
>
> *Computational Latency*.
> We agree that our method is slower than feedforward depth models; however, this trade-off does not diminish our core contribution (L:34-39). We are similar in runtime to other diffusion-based inference-time optimization methods [I], and future advances in this space will also benefit our runtime.
>
> **Weakness 3: Justification for defocus blur**.
> Defocus blur provides metric depth cues and is readily available in images as it's a fundamental property of the camera lens. Compared to other cues, such as stereo (2 cameras), defocus blur allows metric depth estimation with a single camera, and specifically without requiring calibration for the camera extrinsics, which is required for stereo. Moreover, while stereo cameras are widely used, their applicability is limited in compact settings like endoscopy. Other camera physics methods, such as coded apertures, require manufacturing custom hardware. In comparison, defocus cues are more accessible, as they are present in images captured with any standard camera lens. While Metric3D resizes images based on camera focal lengths to ensure stable training over datasets of different depth scales, it doesn’t provide a physically grounded depth cue like defocus blur, and thus cannot be leveraged for per-scene inference time optimization.
>
> *Unique Challenges and benefits of Marigold + defocus blur*.
> Defocus blur offers a differentiable forward model using an AIF and depth map, making it suitable for integration with Marigold at test time. To address mismatches with real cameras (L234, Fig. 2), we adopt a more accurate disk kernel over a Gaussian. While this model simplifies gradient computation, it ignores occlusions, leading to minor errors at depth discontinuities.
>
> **Weakness 4: Marigold Specific Advantages:**.
> 1. Unlike feedforward methods such as DepthAnythingv2 or UniDepth, diffusion-based methods like Marigold provide access to a low-dimensional latent manifold that maps from a noise vector to a relative depth map differentiably. This enables test-time refinement of Marigold’s predictions by incorporating physical cues like defocus blur through differentiable forward models, as shown in Fig. 3.
> 2. Marigold is obtained by fine-tuning StableDiffusion [G] (trained on internet-scale RGB data), allowing it to inherit strong real-world generalization capabilities from StableDiffusion. Most other feedforward models are trained only on RGBD datasets, which are relatively smaller compared to the stable diffusion training dataset.
> 3. Marigold also retains the well-known benefits of diffusion-based depth predictors, such as better fine detail recovery [F] compared to feedforward methods. Additionally, Marigold also provides a latent consistency checkpoint, which speeds up inference compared to using Geowizard.
>
> [A] Depth Estimation via Reconstructing Defocus Image: CVPR 2023.
> [B] Aperture Supervision for Monocular Depth Estimation: CVPR 2018.
> [C] High-Accuracy Stereo Depth Maps Using Structured Light: CVPR 2003.
> [D] Passive snapshot coded aperture dual-pixel RGB-D imaging: CVPR 2024.
> [E] NYU Depth Dataset V2: ECCV 2012.
> [F] GeoWizard: Unleashing the Diffusion Priors for 3D Geometry Estimation from a Single Image: ECCV 2024.
> [G] SDXL: ICLR 2024.
> [H] Bridging Unsupervised and Supervised Depth From Focus via All-in-Focus Supervision: ICCV 2021.
> [I] DITTO: Diffusion Inference-Time T-Optimization for Music Generation : ICML 2024

---

> > ### Comment · Reviewer_1d2m · 2025-08-03
> >
> > I would like to thank the authors for their detailed and well-structured rebuttal, which has effectively clarified many of my initial questions. The new experimental results and justifications are appreciated and have significantly strengthened the paper.
> >
> > However, a few key questions and points of clarification remain. I believe addressing them will further improve the rigor and clarity of the submission.
> >
> > 1. **Regarding the Comparison Baselines (Weakness 1):**
> >
> > * I appreciate the new comparisons provided in Table 2. However, I remain concerned that the baselines, particularly [B], are not directly comparable to the proposed method. A key difference lies in the paradigm: your method relies on training-free, inference-time optimization on a powerful, pre-trained foundation model, whereas [B] is a unsupervised learning approach on a depth estimation with aperture supervision. This makes the comparison less meaningful for establishing the effectiveness of your proposed framework.
> >
> > * A more relevant comparison would involve a different baseline: fine-tuning Marigold on a metric depth dataset and then applying it to a zero-shot scene, compared to your inference-time optimization on that same scene. This would more directly demonstrate the value of your per-scene optimization approach over a globally fine-tuned model as most prior works leverage fine-tuning to achieve metric depth results, but might suffer from domain gap on zero-shot scenes.
> >
> > * Additionally, please provide a proper citation for the "MultiAperture Depth Supervision" baseline for clarity.
> >
> > 2. **Regarding the Ablation Study in Figure 6:**
> >
> > * The explanation of Figure 6 in the paper seems to contradict the clarification provided in the rebuttal. The original paper mentions the degradation of Marigold with a single blurred image, while the rebuttal states this was "not as a baseline but just to check if a single image is sufficient for metric depth estimation." This is confusing. I kindly request that the authors provide a clear and unified statement in the paper to explain the purpose of this ablation study and the conclusions drawn from it.
> >
> > 3. **Regarding the Justification for Defocus Blur (Weakness 3):**
> >
> > * The rebuttal effectively justifies the choice of defocus blur over stereo due to the single-camera requirement. However, a potential weakness of using multi-aperture shots is the phenomenon of 'focus breathing,' where the field of view can subtly change with the aperture. Could the authors please discuss how this potential misalignment is handled or mitigated in the capture process or during the optimization? This would strengthen the claim that the method avoids the need for camera calibration.
> >
> > Minor Fixes:
> >
> > * Please correct the title of paper [A] to its proper name: "Fully Self-Supervised Depth Estimation from Defocus Clue."

---

> ### Author Response · Authors · 2025-08-06
> **Clarifications regarding Figure 6 and focus breathing**
>
> We thank the reviewer for their response and are glad that our rebuttal clarified many of their concerns. Below, we address the follow-up questions on Figure 6 and focus breathing raised by the reviewer in their official comment. Please see our next set of comments for our response to the concerns regarding baseline comparisons raised by 1d2m and MnCo in their official comment.
>
> **Figure 6 clarification:** In our method, Marigold takes in AIF as input (captured at F/22), and predicts a relative depth map. We then synthesize a blurred image, using our forward model (eq 7 in the paper) that uses the all-in-focus (AIF), relative depth from marigold, and the learnable scale-offset parameters. We then compare the synthesized blurred image with the captured blurred image (at F/8), which contains sufficient blur to discern metric depth cues.
>
> Of these two inputs, it is well known [A] that the blurred image provides metric depth cues and thus is an important input. To investigate the impact of the AIF in this setup, we don’t provide the AIF as input, and use only a modestly blurred image (F/16) as input in our setup, for both Marigold as well as for comparison with the synthesized blurred image. We will clarify in the paper that the experiment for Fig. 6 was meant to be an ablation study, to inspect the impact of the AIF image in our setup, and whether our proposed approach can work with a single, modestly blurred image.
>
>  As discussed in the main paper (Fig. 6) and the rebuttal response to 1d2m, we observe that Marigold performs poorly with even modest aperture blur, resulting in significant inaccuracies as compared to using two images. This motivates the two-image approach: the AIF provides high-quality depth from Marigold, and the blurred image provides metric depth cues.
>
> **Focus Breathing:** We would like to clarify that focus breathing is the change in field of view (FoV) caused by adjusting the *focus distance* of the camera lens [C]. This effect arises from variations in optical magnification with focus distance. Methods such as [A], which rely on multiple inputs captured at different focus distances (focal stacks), require alignment to account for these FoV shifts.
>
> Importantly, focus breathing *does not* occur when varying the aperture; it happens *only* when adjusting the focus distance.
>
> By definition [D], the aperture/aperture stop is a surface in the system whose size affects only the area over which light is collected, and not the angle (FoV). Since our method uses inputs captured at different apertures but the *same* focus distance, no FoV alignment is needed. This gives our method an advantage: unlike focal stack-based approaches, we avoid both FoV misalignment and the need for extrinsic calibration. We have also demonstrated competitive performance at different focus distances (0.8m in Table 1 of the main paper and 1.5m in Table 1 in response to MnCO’s review).
>
> [A]: Fully Self-Supervised Depth Estimation from Defocus Clue: CVPR 2023.
> [B]: Aperture Supervision for Monocular Depth Estimation: CVPR 2018.
> [C]: An Implicit Neural Representation for the Image Stack: Depth, All in Focus, and High Dynamic Range: Siggraph Asia 2023.
> [D] Ch 7.1 in Fundamentals of Optics 4th Edition, by Francis A Jenkins, Harvey E. White.

---

> > ### Comment · Reviewer_1d2m · 2025-08-08
> >
> > I would like to thank the authors for their thorough and highly constructive engagement with my review. I have now reviewed all the responses and new table provided in the rebuttal.
> >
> > I am satisfied that you have effectively addressed all of my concerns regarding baseline comparisons, practicality, and the technical justifications behind your method. Your rebuttal has significantly strengthened the paper's claims and rigor.
> >
> > Regarding the discussion on focus breathing, I am satisfied with the technical clarification that focus breathing, as a change in FoV, does not occur with a change in aperture. This is a robust point that strengthens the paper's foundation. My original concern was more rooted in the practical imperfections of real-world lenses, where other effects like focus shift or aperture-dependent geometric distortions may still necessitate some form of alignment for a fully robust implementation. However, I understand that these are minor, real-world considerations and do not diminish the key advantage your method holds over focal-stack approaches. Your response on this topic has been highly convincing.
> >
> > I have no further questions and will be updating my review and scores accordingly.

---

> ### Author Response · Authors · 2025-08-06
> **Regarding baseline Comparisons : Part 1**
>
> **Comparisons with multi-aperture methods**
> 1d2m rightly pointed out in their initial review that our approach relies on multi-aperture inputs, and suggested that we compare our method to previous DfD-like / multi-aperture methods. We added those comparisons in (Table 1, 1d2m response), as they offer valuable context for comparing our method with previous work.
> Baseline [B] in our rebuttal utilizes the same type of multi-aperture inputs as our method. Like [B], our method also doesn’t directly use depth supervision, as our optimization tries to minimize the error between the synthesized and observed blurred image. Given these points, [B] can be considered a fair comparison to our method.
>
> **Comparison with fine-tuning-based methods**.
> As suggested by MnCO and 1d2m in their initial reviews, we provided comparisons with a camera physics + deep learning based method in our rebuttal (Table 1, MnCO response and Table 2, 1d2m response) with 2 baselines.
> Our method relies on a strong relative depth foundation model (RDFM), and incorporates defocus cues at test-time, without re-training the model. Hence, in the initial submission, we did not compare with fine-tuning-based methods, as they are outside the premise of our claims.
>
> While such a comparison would be valuable, a fair comparison would be between our method, i.e., RDFM + test time optimization, and Metric depth models obtained by fine-tuning RDFMs (DepthAnythingv2 [A1] / Marigold) on Metric depth datasets with a restricted depth range (L:225, 290 in paper).  We have not added a comparison with ZoeDepth as it is less performant than DepthAnythingV2 [A1] and MLPro [A2] (which we have already compared with and whose publications already compare to ZoeDepth). As noted by 1d2m in their comment, inference time optimization (our method) can be valuable in zero-shot settings, where fine-tuning-based methods would struggle due to the domain gap. We validate this by evaluating 3 additional fine-tuning-based methods on our collected real dataset in a zero-shot setting. Specifically, we consider the following baselines -
> 1. DepthAnythingv2 fine-tuned on Hypersim (metric depth), as suggested by MnCO. For this baseline, we used the pre-trained model fine-tuned on this configuration released by the authors.
> 2. DepthAnythingv2 fine-tuned on NYU-v2 (metric depth). We used the official fine-tuning code in the DepthAnythingv2 repo to fine-tune the base model for 25 epochs.
> 3. Marigold Fine-Tuned on NYUv2, as suggested by 1d2m. We fine-tuned Marigold for 60 epochs using the official training code. To obtain both depth and RGB latents, Marigold uses the same encoder, requiring inputs normalized to [0,1]. This makes direct fine-tuning with metric depth supervision non-trivial. As a practical proxy, we normalize the GT depth maps (for training supervision) using a fixed global minimum and maximum depth for the full NYUv2 dataset (0.1 m, 10 m), rather than the originally used per-image normalization. Developing optimal fine-tuning strategies for Marigold with true metric depth supervision is beyond the scope of this work; our setup here serves solely as a baseline for comparison and context.
>
> We summarize our results in Table 4 (our dataset) and Table 5 (nyuv2 dataset) in the next comment (Baseline Comparisons Part 2), which validates the utility of our method in zero-shot scenarios, where fine-tuning-based methods can potentially suffer in performance due to the domain gap (Table 4). On our real dataset (Table 4), we outperform all the fine-tuning baselines. On the NYUv2 dataset (Table 5), we either outperform or are comparable to all the baselines except DepthAnythingV2 fine-tuned on NYUv2, which is expected, as the fine-tuning dataset is in the same distribution as the test dataset. Please see the captions of Table 4 and Table 5 in the next comment for more details.
>
> [A1] Depth Anything V2: Neurips 2024.
> [A2] Depth Pro: Sharp Monocular Metric Depth in Less Than a Second: ICLR 2025.
> [B]: Aperture Supervision for Monocular Depth Estimation: CVPR 2018.

---

> ### Author Response · Authors · 2025-08-06
> **Baseline Comparisons: Part 2 -- Comparing with fine-tuning based methods**
>
> | Method        |  $RMSE \downarrow$ |  $REL \downarrow$  | $ \log10 \downarrow$|  $\delta_1 \uparrow$   |  $\delta_2 \uparrow$  | $\delta_3 \uparrow$   |  CD  $\downarrow$  |   FA $\uparrow$   |
> |---------------|--------|-------|-------|-------|-------|-------|--------|--------|
> | Ours, No FT          | 0.273  | 0.125 | 0.052 | 0.879 | 0.975 | 0.991 | 0.103  | 0.870  |
> | Marigold FT NYUv2    | 0.475  | 0.231 | 0.098 | 0.634 | 0.878 | 0.963 | 0.175  | 0.808  |
> | DepthAnyV2 FT HyperSim   | 0.523  | 0.314 | 0.151 | 0.178 | 0.825 | 0.971 | 0.208  | 0.616  |
> | DepthAnyV2 FT NYUv2  | 0.407  | 0.252 | 0.096 | 0.654 | 0.832 | 0.988 | 0.183  | 0.733  |
> **Table 4: Comparing our method with RDFM + Fine-tuning (FT) on metric depth datasets on our real dataset.** Our method (no fine-tuning, row 1) outperforms all the fine-tuning-based methods across all metrics on our real dataset. DepthAnythingV2 shows better performance when fine-tuned on NYU-v2 (row 4)  as compared to fine-tuning on HyperSim. This behavior is expected since NYUv2 (real, maximum depth 10m) would have a lesser domain gap with our collected dataset (real, maximum depth range ~3.5m) compared to HyperSim (synthetic, maximum depth 20 m).
>
>
> | Method | $REL \downarrow$ | $RMSE \downarrow$  |$ \log10 \downarrow$ | $\delta_1 \uparrow$ |  $\delta_2 \uparrow$ | $\delta_3 \uparrow$ |
> |----------------------------------------------------------|-------------------------|-------------|---------|------------|------------|------------|
> | Ours, No FT |  0.150 | 0.535 | 0.045 | 0.899 | 0.949 | 0.969 |
> | Marigold FT NYUv2 | 0.1108  | 0.3730 | 0.0511  | 0.8689 | 0.9834 | 0.9965 |
> | DepthAnythingV2 FT HyperSim | 0.2136 | 0.6548 | 0.0822  | 0.6824 | 0.9685 | 0.9930  |
> | DepthAnythingV2 FT NYUv2 | 0.0790 | 0.2990 | 0.0340  | 0.9510 | 0.9930 | 0.9980  |
> **Table 5: Comparing our method: Rel Depth Foundation Model (RDFM)  + test time optimization with other baselines: RDFM + Fine-tuning on metric depth datasets on the NYUv2 test set.** Our method with no fine-tuning (FT) outperforms DepthAnythingV2 FT on HyperSim (row 3), and is comparable to Marigold FT on NYUv2 (row 2). This highlights that fine-tuning-based methods can be sensitive to the domain gap between the FT dataset and the target test set, as DepthAnything only improves when fine-tuned on NYUv2  (row 4). We’d like to note that it is well established [A3, A4] that a training-based method, when trained on the *same dataset*, can be better than inference time methods. Our results show the key advantages of inference time methods: trading speed / slight performance for increased adaptability and no training.
>
> [A1] Depth Anything V2: Neurips 2024.
> [A3] DITTO: Diffusion Inference Time T Optimization: ICML 2024.
> [A4] End-to-End Diffusion Latent Optimization Improves Classifier Guidance: ICCV 2023.

---

### Official Review · Reviewer_W1fb · 2025-07-03

**Clarity:** 3
**Significance:** 3
**Originality:** 4
**Rating:** 5
**Confidence:** 5

**Summary:**

This paper presents a novel method for upgrading a relative depth map to metric via unsupervised test-time optimization that requires the existence of a second, defocused image of the scene besides the primary sharp image. A major contribution of the paper is the introduction of a rigorous, physics-based defocus blur forward model, which depends on the predicted metric depth map and allows to define a self-supervised loss based on consistency of the captured blurry image and the respective blurry image which is synthesized based on the all-in-focus image, the relative depth map, and the learned scale and shift. The authors validate their approach on a small real-world indoor dataset they have collected and demonstrate superior performance compared to state-of-the-art direct learned metric depth estimators on their dataset.

**Questions:**

Please refer to the questions which are included in the lists of Major Weaknesses and Minor Weaknesses in my response above.

**Ethical Concerns:**

["NO or VERY MINOR ethics concerns only"]

**Final Justification:**

The paper presents an innovative, optically and geometrically rigorous avenue towards metric depth estimation and has shown, also taking the rebuttal into account, very convincing results. In the rebuttal, the authors have clearly addressed the initial point on applicability/practicality of the method, the flexibility to multiple diffusion-based frameworks, the accuracy of their 3D predictions, the merit of the method against previous learned multi-shot / multi-aperture methods, and the quality of depth ground truth and alignment of it with the camera coordinate frame. Thus, I have increased my rating to 5: Accept, and I highly recommend acceptance of the paper.

**Limitations:**

Yes.

**Paper Formatting Concerns:**

None.

**Quality:**

3

**Strengths And Weaknesses:**

Strengths:

1. The usage of the central defocus cue for depth in a learned end-to-end framework via test-time optimization is innovative and insightful and it provides an interesting alternative to direct learned models for metric depth estimation.

2. The derivation of the complete model, especially the defocus blur forward model, is mathematically solid, thorough, and well-grounded on optics.

3. The experimental results on a small yet real-world dataset are very satisfactory, as the proposed method outperforms existing state-of-the-art direct learned metric depth estimators substantially on this dataset across all evaluation measures.

Major Weaknesses:

1. The applicability of the proposed method is quite limited. As the authors already acknowledge, the employed forward model limits the method to indoor scenes with restricted depth ranges. Moreover, the need to capture two different images with different apertures limits the method to specialized camera types such as DSLR, apparently making it inapplicable to e.g. phone cameras or other commodity cameras. At the same time, the method cannot work for dynamic scenes including even slow object motions, as the two images - the all-in-focus and the defocused one - need to be captured in non-overlapping temporal exposure intervals. Thus, the generality and universality of the method is hurt.

2. Metric depth on its own right is not sufficient for metric estimation, as one also needs the directions of the rays corresponding to the pixel grid in order to obtain a metric 3D estimation of the scene structure. Camera intrinsics thus need to be known or estimated. Supposing that the proposed method were applied to an in-the-wild setting with unknown intrinsic calibration, its output is thus insufficient to achieve the aforementioned goal. While the majority of the monocular metric depth methods assume known camera parameters, there are a few that do not [39, A] and that can estimate both camera intrinsics and scene range/depth. Moreover, the quality of metric estimation needs to be evaluated with 3D-wise metrics, such as the Chamfer distance and the F-score. The restriction to mere metric depth maps living in the 2D image space and to traditional depth metrics limits the practical relevance of the method.

3. The authors state in L. 239-240 that they align the predicted depth map with the ground truth depth map before performing the evaluation. What exactly does this alignment entail? Is it applied for competing methods as well? I note here that a major challenge and desideratum in metric depth estimation is to lift the need for aligning / rescaling the model's output in any way and to rather obtain directly a metrically accurate prediction. This is already satisfied in the standard evaluation of recent metric depth estimation works [39, A]. Can the authors present comparisons without the aforementioned alignment?

Minor Weaknesses:

4. It is unclear what the exact optimization algorithm used at inference (cf. L. 212) is. Is it simple gradient descent? Or does it use conjugate gradients? Or is it of another type? What step sizes are used across the 200 iterations? The authors should clarify these points.

5. The exact dependence of the range of the output depth with the scale and shift parameters $\alpha$ and $\beta$ is unclear to me. Based on Eq. (9), since the minimal values of the sigmoids are 0, the overall output depth value could be as low as 0. However, the authors state in L. 186 that the lower bound of depth is $s_{\text{min}}$, not 0. Moreover, in Fig. 5 (right) and L. 266-269, the scale and shift parameters $\alpha$ and $\beta$ seem to be defined to range between 0 and 1, while based on Eq. (9) they should range between 0 and $s_{\text{max}}$ and $s_{\text{min}}$, respectively. Which of the above is true?

[A] Piccinelli et al. UniK3D: Universal Camera Monocular 3D Estimation. In CVPR, 2025.

---

> ### Author Rebuttal · Authors · 2025-07-29
>
> We thank the reviewer (W1fb) for appreciating our experiments on real-world data and our insights on incorporating defocus cues with pretrained diffusion models, and have addressed their questions below.
>
>
> **Major Weakness 1: Concerns about limited applicability**.
> *1a: Restricted depth range, dynamic scenes*.
> We agree with W1fb that our method is best applicable to scenes with a restricted depth range, which is predominantly true for indoor scenes, but also possible for outdoor scenes (e.g., STAIRS in our dataset). We will clarify this in L:289.
>
> While dynamic scenes are not the focus of our work, we acknowledge that handling dynamic scenes in our framework would be valuable. Our current assumption of static scenes aligns with standard practice in monocular depth estimation, as followed by prior works [A, B, E, D]. Note that a two-exposure imaging method does not preclude application to scenes containing any motion. In principle, short exposures can be used with rapid actuation of a camera aperture to capture our two input images with a delay that is on the order of 2x slower than a single image alone. This would only require the motion to be reduced by 2x vs. a single image method set to capture with minimal motion blur. Admittedly, some inter-exposure alignment would likely be needed, but we leave that to future work.
>
> *1b: Variable aperture cameras are also readily available on commodity devices*.
> Our method only requires a single variable aperture camera, which is readily available in the form of handheld commodity devices like Canon DSLRs (as used in our work) and even in some smartphones such as the Samsung Galaxy S9, S10, and upcoming S26. While fixed aperture cameras are more common, variable aperture is still widely accessible and avoids the need for custom hardware like phase or aperture masks used in prior methods [D, F].
>
> **Major Weakness 2: 3D metrics and availability of camera Intrinsics**.
> *2a: Additional 3D metrics*.
> We agree that 3D point cloud metrics such as Chamfer Distance and F1 score (derived from Chamfer Distance) [A, B] are also representative metrics for 3D metric depth; we have added them to *Table 1* below for our method and the competitive baselines. We will add these numbers in a revision. Our reported metrics (L:234 in main paper) are also well accepted in the literature [B, E] for depth accuracy – $\delta_1, \delta_2, \delta_3, RMSE, log10$ capture metric 3D depth accuracy.
>
> | Method             | RMSE $\downarrow$ | REL $\downarrow$   | log10 $\downarrow$ | δ1 $\uparrow$    | δ2 $\uparrow$    | δ3 $\uparrow$    | CD $\downarrow$    | FA $\uparrow$     |
> |--------------------|-------|-------|-------|-------|-------|-------|-------|--------|
> | Ours                | 0.273 | 0.125 | 0.052 | 0.879 | 0.975 | 0.991 | 0.103 | 0.870  |
> | MLPro [E]             | 0.468 | 0.246 | 0.105 | 0.597 | 0.821 | 0.990 | 0.205 | 0.696  |
> | UniDepth [A]          | 0.644 | 0.376 | 0.157 | 0.259 | 0.684 | 0.954 | 0.244 | 0.633  |
> | Metric3D [B]          | 0.459 | 0.295 | 0.106 | 0.650 | 0.825 | 0.895 | 0.180 | 0.725  |.
>
> **Table 1: Results with 3D Metrics (CD: Chamfer Distance), (FA: Aggregated F1 score as used in UniDepth [A]) on our dataset**. On the newly added 3D metrics CD (chamfer distance) and FA (aggregated F score), our method outperforms the existing baselines on our real dataset. This is consistent with the trends observed with the other metrics as well. $\uparrow, \downarrow$ indicate whether higher or lower is better, respectively.
>
> *2b: Availability of Camera Intrinsics*.
> Our capture setup involves capturing an in-focus and a blurred image from a real DSLR camera. The metadata in these images contains EXIF data (camera intrinsics) by default. As the reviewer noted, this is standard practice, with most methods similarly relying on EXIF metadata [G, H]. Hence, we do not focus on the case of unknown camera intrinsics in our work, as that information is always going to be available by design in our capture setup.
>
> **Major Weakness 3: Processing the Intel RealSense depthmap to prepare our dataset**.
> The input images in our setup are captured with an RGB camera, and the ground truth depth map is captured from a RealSense camera (see Fig. 2c in the paper for the capture setup). These two cameras are physically close but not co-located; as a result, the depth from the RealSense does not geometrically correspond to the RGB camera (due to geometric misalignment).
>
> To handle this misalignment, the cameras need to be calibrated using standard camera calibration. This is a standard data pre-processing step [C, D], and is also done by real public RGBD datasets like NYUv2 [C], which allows them to provide geometrically aligned RGB images and depth maps. This allows a fair comparison between the predicted depth from the RGB camera and the GT depth in the same coordinate system. In L:239-240 of our work, this is what we refer to as alignment.
>
> At a high level, first, the pose between the RGB and RealSense cameras is estimated via standard camera calibration (section C of the supplement, L:241 main paper), similar to [D]. The RGB depth map is warped to the GT depth camera’s frame using the estimated pose, enabling comparison between the predicted and GT depth in a shared geometric space.
>
>  *This alignment is crucial for a principled and accurate comparison between GT depth and predicted depth, and is the only alignment we use in our work – both for our method and the baselines.*  Therefore, all reported metrics in the rebuttal and Table 1 of the main paper are computed directly on the raw metric depth outputs.
>
> **Minor Weakness 1: Exact Inference Algorithm, Learning Rates**.
> We have included these details in section A of the supplement, which we reference in L:214 of the paper. We used the Adam optimizer in PyTorch for our optimization, and section A of the supplement contains learning rates and other optimization details.
>
> **Minor Weakness 2: Dependence of output depth on scale and shift parameters ($\alpha$/$\beta$)**.
>  The scale and shift parameters $\alpha,\beta$ are parameterized such that $\alpha\in[0, s_{max}]$ and $\beta\in[0,s_{min}]$, where $s_{min}, s_{max}$ denote the maximum possible values for the lower and upper bounds of the scene depths. We assume $s_{min}, s_{max}$ to be known since we operate in a restricted depth range. (*L 184:186, main paper*).   To ensure both boundedness, and differentiability during optimization, $\alpha, \beta$ are parameterized as (L:184, L:185 main paper) –
> $$
> \alpha = s_{\text{max}} \cdot \sigma(a)~~~\textrm{and}~~~
> \beta = s_{\text{min}} \cdot \sigma(b),$$ where $a,b$ are the learnable unbounded variables, and $\sigma(\cdot)$ denotes the sigmoid activation.
>
> For the performance ablation plot with $\alpha,\beta$ values (Fig. 5), we mention in the figure caption that the $\alpha,\beta$ values are normalized to 0-1 ($\alpha \rightarrow \frac{\alpha}{s_{max}}, \beta \rightarrow \frac{\beta}{s_{min}}$) for plotting.  We will update the figure caption to clarify this.
> [A] UniDepth: Universal Monocular Metric Depth Estimation. CVPR 2024.
> [B] Metric3D: Towards Zero-shot Metric 3D Prediction from A Single Image: ICCV 2024.
> [C] NYU Depth Dataset V2: ECCV 2012.
> [D] Passive snapshot coded aperture dual-pixel RGB-D imaging: CVPR 2024.
> [E] Depth Pro: Sharp Monocular Metric Depth in Less Than a Second: ICLR 2025.
> [F] Learning Phase Masks for Passive Single View Depth Estimation: ICCP 2019.
> [G] Depth Estimation via Reconstructing Defocus Image: CVPR 2023.
> [H] Aperture Supervision for Monocular Depth Estimation: CVPR 2018.

---

> > ### Comment · Reviewer_W1fb · 2025-08-04
> > **Comment on rebuttal by Reviewer W1fb**
> >
> > Dear authors,
> >
> > Thank you for your detailed and convincing response.
> >
> > **1. Applicability**
> >
> > Your explanation about the capacity of the method to also handle dynamic scenes thanks to the availability of appropriate hardware, despite certain constraints on the magnitude of the involved motion compared to the standard single-aperture case of common depth baselines, has sufficiently justified to me the broad relevance of the approach. Moreover, the explanation on the availability of variable-aperture cameras on commonly used smart phones further strengthens the applicability of the method and largely lifts this initially perceived weakness.
> >
> > **2. 3D metrics and full 3D predictions**
> >
> > It is true that full 3D outputs are possible to obtain from an initial depth map when the ground-truth intrinsics are given, which is the case for most controlled setups, such as the ones considered by the authors. While ideally, for in-the-wild scenarios, one would like to predict a metric 3D point map just from image(s), it is true that only few methods are able to achieve this at the moment. In addition, when given the camera intrinsics, the proposed method achieves compelling performance in 3D metrics, such as Chamfer distance and $F_A$ score, outperforming the previous state of the art in 3D estimation. As a result, while the associated limitation remains, its effect is small and should clearly **not** be decisive for evaluating the submission. Nonetheless, since methods such as UniDepth [A] are compared to the authors' approach as well as to other competing methods in a not perfectly fair manner (as [A] is tasked with predicting both intrinsics and metric scene range), it should be clearly marked in the table comparisons of the paper which methods are given the intrinsics and which not.
> >
> > **3. Depth map alignment**
> >
> > Thank you for the detailed clarification. There was a misunderstanding on my side, as I initially thought that by "alignment" you were referring to alignment of the range of initially predicted metric depth values to the actual range of the ground-truth depth maps. Instead, it is now clear that this alignment actually refers to **extrinsically** aligning the original ground-truth depth map recorded by the RealSense sensor with the slightly different 6D pose of the DSLR camera. This alignment is standard practice for any RGBD dataset, such as NYUv2, and is perfectly legit. Thus, this concern of mine is fully lifted.
> >
> > *M1. Optimization algorithm at inference*
> >
> > Thank you for the clarification, this aspect of the method is clear. Maybe it is a good idea to include some key details, such as the choice of the Adam optimizer, in the revised version of the main paper. In any case, I consider this point addressed.
> >
> > *M2. Scale and shift of output depth*
> >
> > The provided clarification resolves my point. Please do update the caption of Fig. 5 to improve clarity.

---

> > > ### Author Response · Authors · 2025-08-06
> > >
> > > We thank the reviewer for appreciating our response and are pleased that they found it convincing. We are glad that our clarifications have addressed both the minor weaknesses and the major concerns regarding applicability and depth map alignment. Below, we address the comment on the sole remaining weakness from their initial review concerning 3D metrics.
> > >
> > > **3D Metrics:** We thank the reviewer for acknowledging in their comment regarding 3D metrics that “while the associated limitation remains, its effect is small and should clearly not be decisive for evaluating the submission.”
> > > As suggested by the reviewer, we will update Table 1 method descriptions to reflect which methods have been provided with camera intrinsics (UniDepth and Ours), and will update L:244-245 for clarity. Regarding the suggestion on factoring in camera intrinsics for evaluation, we have also evaluated Metric3D with camera-intrinsics information on our dataset. We observed that providing camera intrinsics boosts Metric3D’s performance, but it still falls short of our method across all metrics. We will add these numbers and clarifications to the main paper as well, in Table 1.

---

> > > > ### Comment · Reviewer_W1fb · 2025-08-06
> > > >
> > > > Dear authors,
> > > >
> > > > Thank you for your further response. This addresses my slight remaining concern. Please make sure to get the methods' indicators correct in Table 1 regarding the exploitation of camera intrinsics at test time; to my knowledge, e.g., [A] predicts intrinsics itself and does not require it as input.

---

> > > > > ### Author Response · Authors · 2025-08-07
> > > > >
> > > > > We are glad that our response addresses the additional concerns of the reviewer. As suggested, we will ensure that the correct indicators are used for the methods in Table 1. Additionally, we would like to clarify some details regarding the evaluation of [A] (UniDepth) with camera intrinsics.
> > > > >
> > > > > **Evaluating [A] with Camera Intrinsics**.
> > > > > We agree with W1fb that [A] (UniDepth) does not necessarily require camera intrinsics as inputs, as it predicts them directly from the RGB image input. The model architecture in [A] internally decouples the camera intrinsics prediction from metric depth prediction. This makes it possible to swap the predicted camera intrinsics with GT intrinsics (if available) at test time (see Fig. 3 in [A]). We used the official codebase of [A] for our comparisons, which supports inference with externally provided ground truth camera intrinsics. This ensured that both [A] and our method had access to camera intrinsics for a fair comparison. Please refer to section 3.2 in [A] for architectural details that enable swapping the predicted camera intrinsics with GT camera intrinsics during inference.

---

> > > > > > ### Comment · Reviewer_W1fb · 2025-08-07
> > > > > >
> > > > > > Dear authors,
> > > > > >
> > > > > > Thank you for the explanation, it makes total sense.

---

### Author Response · Authors · 2025-08-09
**Appreciation and Acknowledgment for the Reviewers and Area Chair**

We would like to thank the reviewers and the AC for making the review process insightful and positively engaging. We appreciate the time and effort invested by the reviewers in providing constructive feedback, as well as the thoughtful discussions that helped us further strengthen the paper. The reviews and suggested comparisons were especially valuable in this regard. We are also grateful for the constructive dialogue during the post-rebuttal phase, which helped us address the concerns mentioned by the reviewers.

---

### Decision · Program_Chairs · 2025-09-17

**Decision:**

Accept (spotlight)

**Comment:**

The paper proposes a training-free way to estimate metric depth from a pair of images taken from the same viewpoint with different apertures. To that end, a diffusion-based monocular depth estimator (Marigold) is combined with a physics-based defocus model. The latter enables an upgrade from relative to metric depth via test-time optimisation. The initial reviews were mixed, in particular there were concerns about generalisation (particularly outdoor settings), about the evaluation protocol (one vs. two input views, alignment to ground truth, known vs. estimated  focal length), and about the defocus mechanism (refocussing vs. aperture). During the extensive discussion, the authors were able to dispel those concerns and convince the reviewers, who explicitly acknowledged the candid and helpful responses and converged to a unanimously positive recommendation (2x accept, 2x borderline accept).The AC agrees.